# Is our dynamical understanding of the circulation changes associated with the Antarctic ozone hole sensitive to the choice of reanalysis dataset?

Andrew Orr[1], Hua Lu[1], Patrick Martineau[2], Edwin P. Gerber[3], Gareth J. Marshall[1], and Thomas J. Bracegirdle[1]

[1]Atmosphere, Ice and Climate, British Antarctic Survey, Cambridge, United Kingdom

[2]Japan Agency for Marine-Earth Science and Technology, Yokohama, Japan

[3]Courant Institute of Mathematical Sciences, New York University, New York, NY, United States

*Correspondence to*: Andrew Orr (anmcr@bas.ac.uk)

**Abstract.** This study quantifies differences among four widely used atmospheric reanalysis datasets (ERA5, JRA-55, MERRA-2, and CFSR) in their representation of the dynamical changes induced by springtime polar stratospheric ozone depletion in the Southern Hemisphere from 1980 to 2001. The intercomparison is undertaken as part of the SPARC (Stratosphere–troposphere Processes and their Role in Climate) Reanalysis Intercomparison Project (S-RIP). The reanalyses are generally in good agreement in their representation of the strengthening of the lower stratospheric polar vortex during the austral spring-summer season, associated with reduced radiative heating due to ozone loss, as well as the descent of anomalously strong westerly winds into the troposphere during summer and the subsequent poleward displacement and intensification of the polar front jet. Differences in the trends in zonal wind between the reanalyses are generally small compared to the mean trends. The exception is CFSR, which exhibits greater disagreement compared to the other three reanalysis datasets, with stronger westerly winds in the lower stratosphere in spring and a larger poleward displacement of the tropospheric westerly jet in summer.

The dynamical changes associated with the ozone hole are examined by investigating the momentum budget, and then the eddy heat and momentum fluxes in terms of planetary and synoptic-scale Rossby wave contributions. The dynamical changes are consistently represented across the reanalyses, and support our dynamical understanding of the response of the coupled stratosphere-troposphere system to the ozone hole. Although our results suggest a high degree of consistency across the four reanalysis datasets in the representation of these dynamical changes, there are larger differences in the wave forcing, residual circulation, and eddy propagation changes compared to the zonal wind trends. In particular, there is a noticeable disparity in these trends in CFSR compared to the other three reanalyses, while the best agreement is found between ERA5 and JRA-55. Greater uncertainty in the components of the momentum budget, as opposed to mean circulation, suggests that the zonal wind is better constrained by the assimilation of observations compared to the wave forcing, residual circulation, and eddy momentum and heat fluxes, which are more dependent on the model-based forecasts that can differ between reanalyses.

Looking forward, however, these findings give us confidence that reanalysis datasets can be used to assess changes associated with the ongoing recovery of stratospheric ozone.

## 1 Introduction

Since the early 1980s, the polar stratosphere in the Southern Hemisphere has exhibited substantial cooling of up to 6-10 K during austral spring in response to the Antarctic ozone hole, driven by the reduction in radiative heating by stratospheric ozone (Randal and Wu, 1999; Thompson and Solomon, 2002). The cooling increases the meridional temperature gradient from the middle to high latitudes in the lower stratosphere, which is associated, via thermal wind balance, with a strengthening of the stratospheric polar vortex (Thompson and Solomon, 2002), and a subsequent delay in its springtime breakup (Keeble et al., 2014). In spring, the anomalously strong stratospheric winds propagate from the middle stratosphere (~10 hPa) to the tropopause over the course of approximately a month, followed by a rapid descent through the troposphere in a few days (Thompson and Solomon, 2002).

The anomalous tropospheric circulation persists throughout austral summer and is characterized by a poleward shift of the extratropical westerly jet stream, or polar front jet (Thompson and Solomon, 2002; Polvani et al., 2011). The tropospheric wind anomalies are fairly uniform throughout the troposphere (i.e., mostly barotropic), manifesting themselves at the surface as a shift in the midlatitude westerly winds, and are associated with a positive phase of the Southern Annual Mode (SAM) index (Thompson and Solomon, 2002; Marshall 2003; Arblaster and Meehl, 2006). The more positive SAM index has led to significant impacts on the regional climate of the extratropical Southern Hemisphere (e.g., Gillett et al., 2006; Marshall et al., 2006, 2013; Orr et al., 2004, 2008; van Lipzig et al., 2008; Thompson et al., 2011; Deb et al., 2018). Since the early 2000s, stratospheric ozone has begun to show signs of recovery (Solomon et al., 2016), with the associated circulation trends slightly reversed or paused (Banerjee et al., 2020).

The dynamical basis of the polar front jet (or the SAM index) in the troposphere involves positive feedbacks between the anomalous westerlies and synoptic-scale eddy fluxes of momentum and heat. The stronger westerlies are accompanied by enhanced transient baroclinic eddy generation, which tend to propagate upward and equatorward from their latitudes of generation, resulting in a flux of momentum into the jet (convergence) that plays a major role in maintaining its persistence and mid-latitude variability (Robinson, 1996, 2000; Lorenz and Hartmann, 2001; Hartmann and Lo, 1998; Gerber and Vallis, 2007).

The stratospheric polar vortex is strongly influenced by planetary-scale waves propagating up from the troposphere (Christiansen, 1999; Plumb, 2010), which are largely associated with eddy heat fluxes and play an important role in transferring heat from low to high latitudes. Heating perturbations in the stratosphere, which alter the meridional temperature gradient (and via thermal wind balance the vertical shear of background winds), have been shown to modulate the upward propagation of

planetary waves, with changes at the tropopause key to controlling the amount of wave activity transferred from the troposphere

into the stratosphere (Matsuno, 1970; Chen and Robinson, 1992; Scott and Polvani, 2006; Martineau et al., 2018a). Furthermore, the attendant stratospheric circulation anomalies can propagate downwards to the tropopause, with the stratospheric forcing subsequently able to alter tropospheric SAM by changing the synoptic-scale eddy feedbacks that maintain variations in the polar front jet (Christiansen, 1999, 2001; Kodera and Kuroda, 2002; Limpasuvan et al., 2004; Song and Robinson, 2004; Smith and Scott, 2016). Consequently, a number of studies have suggested that the circulation changes

induced by the ozone hole involve alterations to these dynamical processes, although the exact mechanisms remain uncertain (e.g., Hartmann et al., 2000; Chen and Held, 2007; McLandress et al., 2010, 2011; Harnik et al., 2011, Shaw et al., 2011; Orr et al., 2012, 2013; Hu et al., 2015).

Orr et al. (2012) performed a model-based study focused on a momentum budget analysis within the Transformed Eulerian Mean (TEM) framework. It was used to test the hypothesis that the circulation changes associated with the ozone hole were

caused by changes to wave forcing and eddy feedbacks. They found that the initial radiative cooling associated with the ozone depletion causes a strengthening of the lower-stratospheric winds, which results in a reduction of upward propagating planetary waves from the troposphere into the stratosphere. This causes a reduction in the wave-driven deceleration of the polar vortex (resulting in its acceleration) which initiates a positive feedback process in which fewer planetary waves propagate from the troposphere into the stratosphere, and further drawing the reduction in wave-driven deceleration and associated strengthened

winds downwards to the tropopause. In addition, the confinement of planetary wave activity at high latitudes in the troposphere is necessary for the coupling of tropospheric and stratospheric changes. Increased low-level baroclinicity (and associated baroclinic wave activity) results in changes to the synoptic-scale wave fluxes of momentum and heat in the troposphere, which are necessary for the poleward displacement of the polar front jet. In late spring/early summer, however, the delayed breakup of the stratospheric vortex associated with the ozone hole permits increased upward fluxes of planetary wave activity from the

troposphere into the lower stratosphere at high latitudes, and consequently, stronger wave-driven deceleration (Orr et al., 2012).

Atmospheric reanalysis datasets combine observations with a fixed weather forecast model in an optimal way to construct a 'best' estimate of the state of the atmosphere. They have previously been used to investigate the circulation trends in the stratosphere and troposphere that occur in response to the ozone hole (e.g. Chen and Held, 2007; Harnik et al., 2011; Shaw et al., 2011; Orr et al., 2012, 2013; Banerjee et al., 2020). These studies tend to be based on a single reanalysis dataset, despite

many others being available (Fujiwara et al., 2017). Reanalysis systems assimilate both conventional data (e.g. radiosonde profiles, surface measurements, and aircraft measurements) and satellite data (e.g. infrared and microwave radiances). The availability of satellite data has increased substantially since the appearance of the ozone hole and contributed to major improvements in accuracy (Marshall, 2003; Sterl, 2004), as prior to the 'satellite era' reanalyses are considered unreliable in the high latitudes of the Southern Hemisphere due to the sparseness of conventional observations (e.g., Gerber and Martineau,

2018).

As reanalysis datasets largely use the same available input observations, differences in the technical details of the reanalysis systems means that they may give different results for the same diagnostics (Fujiwara et al., 2017). Key differences in current reanalysis systems include the data assimilation strategy, which include three- and four-dimensional variational (3D-VAR and 4D-VAR) approaches, as well as 3D-FGAT (First Guess at Appropriate Time). 4D-VAR makes better use of observations than either 3D-VAR or 3D-FGAT, resulting in substantial improvements, while the 3D-FGAT approach is regarded as an intermediate step between 3D-VAR and 4D-VAR (Fujiwara et al., 2017). Differences in the forecast models used are also important, as they have their own biases throughout the atmosphere. Therefore, reanalysis datasets do not necessarily agree on how the Southern Hemisphere circulation responds to the ozone hole, possibly making the results reanalysis dependent. This is perhaps especially an issue in the stratosphere, as compared to the troposphere this region is characterised by smaller volumes of observational data available for assimilation and larger biases in observational data (Fujiwara et al., 2017), implying a greater reliance on the performance of the forecast model and its representation of dynamical processes (e.g., Orr et al., 2010). The representation of the underlying dynamics in reanalyses is therefore an additional concern, which has not been examined for the Southern Hemisphere despite showing nonnegligible differences for some diagnostics in the Northern Hemisphere (e.g., Bengtsson et al., 2006; Lu et al., 2015; Martineau et al., 2016, 2018b; Chemke and Polvani, 2020).

The primary aim of this study is to compare trends in the Southern Hemisphere circulation over the 1980 to 2001 period, associated with the ozone hole, in four widely-used reanalyses, and to analyse their connection to changes in key dynamical balances to establish whether they consistently support the proposed mechanisms associated with the ozone hole. The four reanalyses datasets examined are JRA-55 (Kobayashi et al. 2015), MERRA-2 (Gelaro et al., 2017), CFSR (Saha et al., 2010, 2014), and ERA5 (Hersbach et al., 2020).

## 2 Data and methods

Details of the four reanalysis systems examined are given in Table 1. See also Fujiwara et al. (2017) for a summary of each of the reanalysis systems. Key differences are that both ERA5 and JRA-55 employ a 4D-VAR (Kobayashi et al. 2015; Hersbach et al., 2020) scheme, while CFSR employs a 3D-VAR scheme (Saha et al., 2014) and MERRA-2 employs a 3D-FGAT scheme (Lawless, 2010). There is a considerable difference in the release date of the forecast model used by each system, which is relevant as the models will have improved over time. ERA5 and MERRA-2 use considerably more recent model versions (year 2016 and 2015, respectively) compared to JRA-55 and CFSR (year 2009 and 2007, respectively). The four systems have a broadly similar horizontal grid spacing of 0.5° or better. The vertical resolution of JRA-55, MERRA-2, and CFSR is broadly similar with approximately 60-70 levels from the surface up to around 0.1 hPa, whereas ERA5 uses 137 levels from the surface up to 0.01 hPa. All the reanalyses assimilate satellite measurements of ozone, although the way that this is treated and the data used varies considerably between reanalysis systems (Davis et al., 2017).

The data used in this study are described by Martineau et al. (2018c) and were produced as part of the Stratosphere-troposphere Processes And their Role in Climate (SPARC) Reanalysis Intercomparison Project (S-RIP) to facilitate the comparison of reanalysis datasets (Gerber et al., in press). They include zonally averaged atmospheric diagnostics of basic dynamical variables and more advanced wave forcing quantities computed using the four reanalyses datasets examined. The variables are
provided every six hours and prepared using a common 2.5° × 2.5° grid and standard pressure levels. For this investigation a subset of the data was retrieved, comprising the period from 1980 to 2001 and 15 pressure levels (1000, 925, 850, 700, 600, 500, 400, 300, 250, 200, 150, 100, 70, 50, 30 hPa). Note that no data is provided for MERRA-2 in the range 1000-500 hPa since, unlike the other reanalyses, it does not provide data extrapolated below the surface. Additionally, ERA5 exhibits a pronounced cold bias in lower stratospheric temperature from 2000 to 2006 due to the use of inappropriate background error
covariances (Simmons et al., 2020). This issue was fixed in a new set of ERA5 reanalysis from 2000 to 2006, termed ERA5.1 (Simmons et al., 2020), which we used instead of ERA5 for this period (hereinafter this combined dataset is referred to as ERA5 for simplicity).

The key variables examined in this study are the trends in the zonally averaged zonal wind $\bar{u}$, the eddy momentum flux $\overline{u'v'}$, and the eddy heat flux $\overline{v'T'}$. Here $T$ is the temperature, $u$ the zonal wind, $v$ the meridional wind, overbars denote zonal averages,
and primes denote deviations from the zonal average. In the Southern Hemisphere, positive (negative) anomalies of the eddy heat flux indicate reduced (enhanced) poleward heat transfer, while positive (negative) anomalies of the eddy momentum flux indicate reduced (enhanced) poleward momentum transfer.

The trends in the individual terms of the momentum budget are also examined using the quasi-geostrophic (QG) form of the TEM momentum equation (Edmon et al., 1980), which is expressed as

$$140 \quad \frac{\partial \bar{u}}{\partial t} = f\bar{v}^* + \frac{1}{a\cos\phi}\boldsymbol{\nabla} \cdot F^{QG} + \bar{\epsilon} \tag{1}$$

where $t$ is time, $f$ is the Coriolis frequency, $\bar{v}^*$ is the residual meridional circulation, $F^{QG}$ is the Eliassen-Palm (EP) flux, $a$ is the mean radius of the Earth, $\phi$ is the latitude, and $\bar{\epsilon}$ represents any residual tendencies. The term on the left-hand-side of Eq. (1) is the zonal wind tendency (i.e., the time derivative). The first term on the right-hand-side of Eq. (1) is the Coriolis torque associated with the residual meridional circulation, an estimation of the net transport of mass which includes both the Eulerian
mean meridional flow and eddy transport (Edmon et al., 1980). The second term on the right-hand-side of Eq. (1) is the wave forcing, represented by the EP flux divergence (hereinafter EPFD). The EP flux takes the form

$$F^{QG} = \left\{ F_\phi{}^{QG}, F_p{}^{QG} \right\} = a\cos\phi \left\{ -\overline{u'v'}, \frac{\overline{v'\theta'}}{\partial\bar{\theta}/\partial p} f \right\} \tag{2}$$

where $\theta$ is the potential temperature, $p$ is pressure, and $F_{\emptyset}^{QG}$ and $F_p^{QG}$ are respectively the horizontal and vertical components of the EP flux (Martineau et al., 2018c). The eddy heat fluxes play a key role in the vertical component of EP flux, which is a measure of the upward fluxes of Rossby wave activity (Edmon et al., 1980).

The key variables examined in Eq. (1) are the trends in the wave forcing (EPFD), and the Coriolis torque $f\bar{v}^*$. Our use of the QG TEM approximation for the momentum budget and the lack of complete access to all the relevant data, preclude us from a meaningful analysis of the trends in the residual term in the momentum budget, so therefore this term is not considered. The residual includes both parameterized gravity wave drag (e.g., Lott and Miller, 1997; Orr et al., 2010) and reanalysis increments in the momentum budget, but also ageostrophic terms and any numerical biases in the model (which therefore cannot be separated as they are all included in a single term). The TEM framework is ideal as a diagnostic tool for identifying the dominant balance between the Coriolis torque on the net poleward transport of mass (quantified by the residual circulation) and the transport of momentum by Rossby waves (quantified by the EPFD term), i.e., examining how changes in these two terms relate to changes in the zonal mean wind, which is therefore the focus of this work. On seasonal time-scales, the EPFD and Coriolis torque terms are the leading order balance in the system: momentum transfer in the free atmosphere is controlled dynamically via eddy heat and momentum transfer (Palmer, 1981).

The results compare linear trends over a 22-year period from 1980 to 2001 for the four reanalyses, focusing on austral spring (September-October-November; SON) and summer (December-January-February; DJF). Note that the magnitude of the trends are often small in comparison to the mean values. Statistical significance testing of the trends is established using the two-sided Student's $t$ test, with a confidence interval of 5%. Statistical significance testing of the differences in the trends between the reanalyses was also tested, but found to be not significant at the 5% significance level. The 1980 to 2001 period was chosen because the time-series of the ozone mass deficit measure of Huck et al. (2007) revealed that the ozone hole first emerged around 1980, with its size steadily increasing until around 2000/2001. So selecting the 1980 to 2001 period maximises trends in circulation and related dynamical quantities (Banerjee et al., 2020). It also provides a clean case study for reanalysis data inter-comparison in terms of atmospheric trends and the associated dynamical connection between the troposphere and the stratosphere in the Southern Hemisphere.

Results based on meridionally-averaged values of the zonal wind, eddy momentum flux $\overline{u'v'}$, eddy heat flux $\overline{v'T'}$, wave forcing (EPFD), and Coriolis torque $f\bar{v}^*$ are areally weighted using the cosine of latitude. This weighting is akin to comparing the angular momentum in the case of zonal wind and the net torques in the case of forcing terms, except for a factor representing the radius of the Earth (which we omit so that the units are more easily interpretable). In the case of eddy fluxes, the weighting accounts for the fact that for equal zonally-averaged fluxes, the associated wave activity fluxes are larger towards the Equator due to increasing latitude circle (Eq. 2). To better compare differences between the reanalyses, ERA5 is chosen as a reference dataset and differences between it and MERRA-2, JRA-55 and CFSR are calculated. The choice of ERA5 as a reference is

somewhat arbitrary, in that we have no a priori expectation that it is closer to the truth. It is, however, the most recently developed reanalysis (Table 1). Furthermore, the vertically integrated momentum flux and heat flux are also computed (Held and Phillipps, 1993) for all waves, as well as planetary (zonal wavenumbers 1-3) and synoptic (zonal wavenumbers 4 and higher) waves. This is to investigate differences in the propagation of synoptic waves in the troposphere and planetary waves in the stratosphere and their contributions to the total wave fluxes. Finally, the trend in the final warming date of the Antarctic polar vortex was calculated using the method of Black and McDaniel (2007), which identified the final warming date as the final time that the zonally averaged wind at 60°S and 50 hPa falls below 10 m s$^{-1}$.

## 3 Results

Since Southern Hemisphere winds have undergone such large changes in response to the ozone hole, we first compare the trends in the zonal wind among the reanalysis datasets. Figure 1 shows height (in pressure coordinates) versus latitude cross-sections of DJF trends in zonally averaged zonal wind for the period 1980 to 2001 for the four reanalyses. All four reanalyses show the expected stronger westerly winds in both the stratosphere and troposphere, with the trends statistically significant. For ERA5, the strongest increases in both the troposphere (up to 2 m s$^{-1}$ dec$^{-1}$) and stratosphere (2.5 m s$^{-1}$ dec$^{-1}$) are confined to a relatively narrow latitudinal band of 55-65°S, although in the stratosphere the enhanced westerly winds expand equatorward to 30°S and poleward to 80°S, which is consistent with an overall strengthening of the climatological polar vortex. In the troposphere the strengthened westerly winds form part of a dipole pattern, with weaker easterly winds at around 40°S, in accordance with a poleward shift of the polar front jet. The results for ERA5, JRA-55, and MERRA-2 are largely in good agreement both in the magnitude of the trends and the statistical significance. For CFSR, the peak wind increase in the stratosphere exceeds 3.0 m s$^{-1}$ dec$^{-1}$, and in both the stratosphere and troposphere the region of maximum increase in the westerlies is located in the range 60-70°S. This is further poleward and larger in magnitude in comparison with ERA5, resulting in the positive differences at 60-70°S and negative differences at 50-60°S when compared to ERA5. Note that CFSR also disagrees with the other three reanalyses in terms of the corresponding (DJF) trends in temperature, evident by enhanced warming below 300 hPa (by ~0.4 K dec$^{-1}$) and cooling between 300 and 100 hPa (by ~-1 K dec$^{-1}$) relative to ERA5, resulting in comparative weakening of the stability near the tropopause (Figure A1).

Following Thompson and Solomon (2002), Figure 2 shows the corresponding time-height cross-sections of the trends in zonally averaged zonal wind (averaged over 50-70°S) from September to February. The expected strengthening of the winds and their descent from the lower stratosphere into the troposphere is apparent in all four reanalyses. For ERA5, the trends in zonal wind can be separated into four stages: i) stronger westerly winds appearing in the lower stratosphere in September, ii) continued strengthening of the lower stratospheric winds from September to early December (peaking at 4 m s$^{-1}$ dec$^{-1}$) and descent to the tropopause, iii) weakening of the anomalously strong westerly winds in the lower stratosphere from December to January and descent of the winds to the surface, and iv) a continued weakening of the anomalously strong stratospheric

winds from December to February, consistent with a delayed breakup of the vortex in summer. According to Orr et al. (2012), these four stages refer respectively to the 'onset', 'growth', 'decline', and 'decay' stages of the lifecycle of the zonal wind response to the ozone hole.

The results for ERA5, JRA-55, and MERRA-2 are again largely in good agreement (with differences not exceeding ±0.6 m s$^{-1}$ dec$^{-1}$). The largest differences among the reanalyses are again associated with CFSR, which shows much stronger stratospheric winds than ERA5 between September and November (i.e., the 'onset' and 'growth' stages), suggesting the initial strengthening of the winds occurs earlier in CFSR. Furthermore, the four reanalyses generally show a similarly delayed breakup of the polar vortex. The final warming date for all reanalyses occurs around 0.9 days later per year or around 19 days later over the period 1980 to 2001 (not shown).

In ERA5 the corresponding time-height cross-section of trends in zonally averaged temperature (Figure A1) demonstrates that the stratospheric cooling associated with the ozone hole lasts from October to January, with a peak of -4 K dec$^{-1}$ in November (which is statistically significant). This agrees with radiosonde observations from Antarctica (Thomson and Solomon, 2002), and is also in agreement with MERRA-2 and JRA-55 results. However, CFSR again contrasts with the other three reanalyses in terms of the temperature trend, evident by both an earlier onset to the cooling (beginning from September) and enhanced cooling between 300 and 100 hPa (by ~-1 K dec$^{-1}$) throughout September to February (also statistically significant).

To further investigate the response of the tropospheric polar front jet during DJF, Figure 3 shows the latitudinal profile of the trend and climatology of the 500 hPa zonally averaged zonal wind for the four reanalyses. The climatologies are nearly identical except poleward of ~70°S and show that the peak winds associated with the jet occur around 50°S. The lack of agreement poleward of ~70°S may be due to a lack of observations over the continent and/or the increase in uncertainty of zonal mean quantities near the pole, an effect of spherical geometry. Positive significant trends (~1.5 m s$^{-1}$ dec$^{-1}$) are found on the poleward flank of the jet while negative trends (~-0.8 m s$^{-1}$ dec$^{-1}$) occur at ~38°S, which is consistent with the results of Figure 1, i.e. a strengthened and poleward shift of the polar front jet in the troposphere. In comparison with other reanalyses, there is a clear poleward shift of ~4° for the CFSR trend, which is also consistent with the stronger poleward shift in the jet shown in Figure 1. The good agreement between the climatological results suggests that the differences in the trends are not due to biases/differences in the climatological strength or location of the tropospheric westerly jet. Note that the anomalous CFSR trend in the polar front jet compared to the other reanalyses is even more apparent at 850 hPa (Figure A2), i.e., near the surface and consistent with fairly uniform (barotropic) wind trend anomalies throughout the troposphere.

**3.1 A dynamical analysis of trends: Balance between EP flux divergence and Coriolis torque**

To study the spread among the four reanalyses in terms of wave driving, Figure 4 (a,d,g,j) shows time-height cross-sections of the trend in EPFD (averaged over 40-80°S) from September to February. For ERA5, in the lower stratosphere the EPFD shows

a statistically significant positive trend during November (i.e., weaker wave drag, coinciding with the 'growth' stage and the peak increase in stratospheric winds), followed by a significant negative trend during DJF (i.e. stronger wave drag, coinciding with the 'decay' and 'decline' stages and a weakening of the strengthened vortex and a delay in its breakdown). This is in dynamical agreement with the temporal evolution of the zonal wind trends in Figure 2 but does not necessarily indicate causality. The total zonal wind acceleration (in the absence of e.g. unresolved small-scale forcing) is largely a balance between the Coriolis torque on the residual meridional circulation and the wave drag on these time scales (Eq. 1). For September and October, the trend in lower stratospheric EPFD is largely negligible, suggesting that the circulation response during this time is primarily radiatively controlled. Both positive and negative trends in EPFD descend from 30 hPa to 300 hPa, indicating a downward influence from the stratosphere. In the lower stratosphere the trend in EPFD shows little difference among the four reanalyses.

Orr et al. (2012) also describe a switch from weaker (in November) to stronger (in DJF) wave drag in response to the ozone hole. They emphasize two factors, i) a positive feedback process whereby an initial strengthening of the polar vortex winds in response to radiative cooling (during the 'onset' phase) plays an important role in conditioning the polar vortex so that fewer planetary waves can propagate up from the troposphere into the stratosphere, resulting in reduced wave drag (during the 'growth' phase): this agrees with the conclusion of Chen and Robinson (1992), that enhanced vertical wind shear at the tropopause is key to reducing the propagation of planetary wave activity into the stratosphere. And ii), a negative feedback process whereby the prolonged existence of the westerly winds due to the delayed breakdown of the stratospheric vortex permits increased upward wave propagation into the stratosphere, resulting in stronger wave drag (during the 'decline' and 'decay' stages).

In the troposphere, EPFD shows bands of negative (positive) significant trends in the upper (middle) troposphere for ERA5 from September through to February (cf. Figure 4a). The agreement among the four reanalyses is poor, with the deviations relative to ERA5 marked by alternating negative and positive horizontal banding, which can be greater in amplitude than the mean trends, and are most prominent for CFSR (e.g., during October). However, the rather large spread in the tropospheric EPFD trends (Figure 4 (a,d,g,j)) are accompanied by relatively small differences in the tropospheric wind trends (Figure 2). Although there is also no evidence of vertically alternating differences in the wind trend (Figure 2), alternating negative and positive horizontal bands are apparent in the temperature trends, albeit located at different levels from the EPFD results (Figure A1).

These results suggest that in the troposphere, the resolved EPFD trend is less directly linked to observations compared to the trends in the zonal wind, and so more forecast model dependent. In addition, the tropospheric circulation is relatively more constrained by observational input in comparison to the stratospheric circulation (Martineau et al., 2016). Lu et al. (2015) found similar alternating stripes in the EPFD when they compared wave driving between ERA-Interim and ERA-40 reanalyses. They showed that one of the main contributors to the EPFD differences was the vertical derivative of the temperature. Note

that interpolation from model levels to standard pressure surfaces could also play a role in discrepancies of the EPFD term, as derivatives are very sensitive to interpolation. Differences in trends in the upward component of the EP flux (Eq. 2), which also includes the vertical derivative of temperature, are also characterized by alternating negative and positive horizontal bands
(not shown).

Figure 4 (b,f,j,n) shows analogous results to the EPFD trends but for the Coriolis torque $f\bar{v}^*$. The trends in this term are typically similar in magnitude to the trends in the EPFD term but of opposite sign. Note that differences in the trends in the Coriolis torque were also of similar magnitude but opposite sign to the differences in the EPFD trends. This is the dominant balance expected under quasi-geostrophic scaling, in part reflecting the fact that both the Coriolis torque on the residual
circulation and momentum flux divergence are dominated by the same term: the partial derivative of $F_p{}^{QG}$ (see Eq. (2)) with respect to pressure, which can be interpreted both as the Coriolis force acting on the net transport of mass by eddies (in the first term on the right hand side of Eq. (1)) and the transport of momentum by eddies, associated with form drag (the second term on the right hand side of Eq. (1); see Vallis (2017), Chapter 10 for further details). Orr et al., (2012) also found that these two terms were of similar magnitude and opposite sign, and that the sum of these two terms agreed well with the zonal wind
tendency.

**3.2 A dynamical analysis of trends: Eddy heat and momentum fluxes**

Figure 4 (c,g,k,o) shows the time-height cross-sections of the trend in zonally averaged eddy heat flux $\overline{v'T'}$ for the four reanalyses. The ERA5 results show a region of significant positive trend in the lower stratosphere in November indicating reduced poleward eddy heat flux / upward wave activity into the stratosphere, which corresponds to the positive trends in
EPFD, i.e., reduced EP flux convergence. Comparison with Figure 2 reveals that this period is also contemporaneous with the descent of the anomalously strong westerly winds / increased vertical wind shear to the tropopause. For DJF, the ERA5 results show a significant negative trend in the lower stratosphere signifying enhanced poleward eddy heat flux / upward propagating wave activity into the stratosphere, which corresponds to negative trends in EPFD, i.e., increased EP flux convergence. For September and October, the trend in lower stratospheric eddy heat flux is much smaller and noisier, but still statistically
significant. This corresponds to the switch from weaker (in November, during the 'growth' stage) to stronger (in DJF, during the 'decline' and 'decay' stages) wave activity propagating into the lower stratosphere described by Orr et al. (2012). The other reanalyses exhibit minor differences compared to ERA5, except for CFSR, which exhibits a stronger negative trend of the eddy heat flux in DJF (and September and October) and a weaker positive trend in November. Additionally, in ERA5 the region of positive trend in heat flux in November appears to start from around the tropopause and extends upward quickly in
time, while this effect is less apparent or more barotropic in the other three reanalyses. Negligible trends in the heat flux can be detected in the troposphere, confirming that changes in the upward propagating waves are confined in the stratosphere (Orr et al., 2012).

Figure 4 (d,h,l,p) shows the time-height section of the trend in zonally averaged eddy momentum fluxes $\overline{u^{'} v^{'}}$ for the four reanalyses. For ERA5, a significant negative trend is found to dominate the lower stratosphere from October to November, indicating enhanced poleward momentum transfer. Hartmann et al. (2000) argued that the enhanced vortex winds / vertical shear in the polar lower stratosphere associated with the ozone hole cause enhanced equatorward propagation of planetary waves, thus more negative $\overline{u^{'} v^{'}}$ in the Southern Hemisphere (i.e., poleward momentum transfer). For the other three reanalyses, the negative stratospheric trend is stronger compared to ERA5, especially in CFSR (consistent with its stronger vortex winds from September to December (Figure 2), which favors increased equatorward wave propagation in the lower stratosphere).

In the troposphere, in ERA5 the trend in eddy momentum flux is marked by persistent negative values from December to February (but not significant), indicating enhanced poleward momentum transfer. This occurs at the same time as the poleward displacement of the polar front jet and anomalously strong westerlies in the troposphere (Figures 1 and 2). This negative trend in eddy momentum flux in the troposphere is evident for all four reanalyses products, although JRA-55, MERRA-2, and CFSR have weaker trends than ERA5. Orr et al. (2012) similarly describe strengthened equatorward synoptic-scale wave propagation in the troposphere in response to the ozone hole during the 'decline' and 'decay' stages. They show that this coincides with enhanced baroclinity at the surface (i.e., an increase in upward propagating synoptic-scale waves) at the same latitude as the strengthened polar front jet. This suggests that the circulation trends are the result of the interactions between the zonal-mean flow and the eddies, which maintain anomalies in the polar front jet / tropospheric annular mode. The flux of momentum into the jet (convergence) balances anomalous surface wind stress associated with the shift (see also Hartmann et al., 2000).

The analysis in the next two sub-sections further explores the differences in the trends in eddy heat and momentum fluxes for November (Figures 5 and 6) and DJF (Figures 7 and 8). The reason for focusing on these two periods is to further examine the switch from weaker (in November) to stronger (in DJF) wave activity propagating into the lower stratosphere, as well as the strengthening and poleward-displacement of the polar front jet in the troposphere (in DJF).

### 3.3 A dynamical analysis of trends: November

Figure 5a shows the latitude-height profile of the zonally averaged eddy heat flux $\overline{v^{'} T^{'}}$ climatology from ERA5 for November, which is dominated by negative values from 45-80°S in the lower stratosphere, consistent with upward propagating waves along the polar vortex edge. Quantitatively similar results can be obtained from the other three reanalyses (not shown). Figure 5 (c,e,g,i) shows the trend in eddy heat flux for November, which for all four reanalyses is marked by significant positive values in the lower stratosphere at 40-80°S, so in agreement with Figure 4 and confirming the reduction of poleward eddy heat flux / upward wave activity flux from the troposphere into the lower stratosphere. Overall, in terms of both magnitude and location, the best agreement is found between ERA5 and JRA-55, while the positive trend in CFSR is around half that of ERA5, indicating a much weaker reduction in upward wave activity from below for CFSR. This is despite CFSR showing

stronger positive wind trends in the lower stratosphere compared to the other reanalyses in November (Figure 2), which is dynamically inconsistent as this would be expected to be associated with a relative stronger (not weaker) reduction in upward wave activity. Figure 6 shows the 100-30 hPa vertically integrated trend (and climatology) of eddy heat flux for all waves, as well as planetary and synoptic waves, again for the month of November. This analysis confirms that the reduced upward wave fluxes in the lower stratosphere are composed of planetary waves, in good agreement with Orr et al. (2012). However, there is a noticeable amount of disagreement in the CFSR trends compared to the other three reanalyses in terms of both amplitude and latitudinal extent. Note that although the vertically integrated trends are not significant at the 5% significance level, they are generally consistent with what is expected from the dynamical argument.

Figure 5b shows the climatology of the November, zonally averaged eddy momentum flux $\overline{u' v'}$ derived from ERA5, which is dominated by negative values at 30-60°S in the lower stratosphere, indicating poleward momentum fluxes. In the troposphere, the climatology is marked by negative values at 30-55°S and relatively smaller positive values at 60-80°S, indicating momentum convergence in mid-latitudes. Figure 5 (d,f,h,j) shows the trend in eddy momentum flux, which for all four reanalyses at around 50-80°S is marked by negative values at ~100 hPa (but not significant), so in agreement with Figure 4 and confirming enhanced poleward eddy momentum flux / equatorward propagation of wave activity. All four reanalyses show this feature, except that the magnitude of the trend is larger in MERRA-2 and even larger and more poleward in CFSR. Note that there are also positive trends at ~300 hPa, which are also apparent in Figure 4. Figure 6 (b,d,f) shows that the negative lower stratospheric trends displayed in Figure 5 are dominated by the contribution from planetary waves. Similar to the eddy heat fluxes, there is a noticeable disagreement between the CFSR trends and the other three reanalyses, while the best agreement is found between ERA5 and JRA-55. Again, although these vertically integrated trends are not significant, they are consistent with the expected dynamical argument.

### 3.4 A dynamical analysis of trends: Austral summer

Figure 7 is analogous to Figure 5, but for DJF. The eddy heat flux $\overline{v' T'}$ climatology for DJF from ERA5 (Figure 7a) is dominated by negative values at 40-60°S, 100-1000 hPa, indicating that upward propagating baroclinic waves are confined largely to the troposphere, as expected in austral summer (Plumb, 2011). Quantitatively similar results can be obtained from the other three reanalyses, with differences of only around 1 m s$^{-1}$ K at a few locations (not shown). Figure 7 (c,e,g,i) shows results for the DJF trend for ERA5, JRA-55, MERRA-2, and CFSR. For all four reanalyses there is a significant negative trend poleward at around 50°S in the lower stratosphere (and to a lesser extent the uppermost region of the troposphere), so in agreement with Figure 4 and confirming the importance of enhanced upward wave fluxes at high latitudes into the lower stratosphere in the summer months (Orr et al., 2012). ERA5 and JRA-55 again show the best agreement, with MERRA-2 and especially CFSR showing larger negative values in the lower stratosphere (~300 hPa).

Figure 8 is analogous to Figure 6, but for DJF and the height range of 30-300 hPa for the eddy heat flux and 100-500 hPa for the eddy momentum flux. The reason for selecting different ranges for the vertical integration was because the strongest trends in eddy heat flux are found from 30-300 hPa for all four reanalyses, and from 100-500 hPa for the eddy momentum flux (Figure 7). Figure 8 (a,c,e) shows that the eddy heat flux trend from 30-300 hPa due to all waves is dominated by statistically significant negative values at 45-80°S, which is poleward of the climatological values at 30-70°S (cf. Figure 7). In agreement with Orr et al. (2012), these trends are dominated by planetary waves at 55-80°S, while synoptic waves also have some role at 45-70°S (both these trends are statistically significant). As the climatological tropopause height is above 300 hPa equatorward of 60°S (Figure 7(a,b)), some of the synoptic waves in this region are actually in the upper troposphere and not the lower stratosphere. Again, ERA5 and JRA-55 are in good agreement, while the MERRA-2 and CFSR trends are both stronger and more poleward. Also, the differences among the four reanalyses are not statistically significant in the latitude bands where statistically-significant trends are detected.

Figure 7b shows the DJF eddy momentum flux climatology from ERA5. The climatology is marked by positive values at 60-75°S, 200-500 hPa and negative values at 30-55°S, 100-500 hPa, so confined largely to the troposphere. Similar climatologies can be obtained from the other three reanalyses with differences of no more than 4 m$^{-2}$ s$^{-2}$ at a few locations within the positive and negative regions shown for ERA5. Figure 7(d,f,h,j) shows DJF trends in momentum flux derived from ERA5, JRA-55, MERRA-2, and CFSR. The trends are marked by significant negative values reaching -5 m$^{-2}$ s$^{-2}$ dec$^{-1}$ in the troposphere at 40-70°S, so consistent with Figure 4 and confirming the importance of enhanced poleward eddy momentum fluxes at the core of the climatological polar front jet in the troposphere (Orr et al., 2012). All four reanalyses capture this feature, except that the magnitude of the trend is largest in ERA5 and ir is slightly shifted in the other three reanalyses. Again, the differences among the four reanalyses are not statistically significant in the latitude bands where statistically-significant trends are detected. This implies that the trends in both heat and momentum fluxes in the stratosphere are robustly captured by all four reanalysis data sets.

Figure 8 (b,d,f) shows vertically integrated results for DJF from 100-500 hPa for the eddy momentum flux. The trends in eddy momentum fluxes due to all waves are also dominated by statistically significant negative values centered at 40-70°S (cf. Figure 7), which is poleward of the climatological minimum values and also dominated by the (statistically significant) contribution from synoptic-scale waves. This is again in agreement with Orr et al. (2012). The four reanalyses, however, exhibit more considerable disagreement in the trends, which are more pronounced than the differences in their climatological values.

**3.5 Sensitivity of the trends to time period**

To further assess the statistical robustness of the trends, we explore the impact of small shifts in the time period of the analysis on the trend. Figure 9 shows time-height cross-sections of the trends in zonally averaged zonal wind for the reanalyses from

September to February for three different 20-year periods (1980 to 1999, 1981 to 2000, and 1982 to 2001) that overlap our analysis period of 1980 to 2001. The trends and spread in zonal wind between the reanalyses for the different periods agree with the results for the 1980 to 2001 period. To examine the robustness of the trends in dynamical quantities, Figure 10 compares the spread of the November trends in 30 to 100 hPa vertically integrated eddy heat flux for the three 20-year periods (c.f. Figure 6). The spread of the trends in eddy heat flux for the different periods are similar, and consistent with the results for the 1980 to 2001 period. Examination of the sensitivity of the trends for the other dynamical quantities examined in this study to the different time periods exhibited a similar robustness (not shown). The differences among the reanalyses are of similar magnitude compared to the sampling uncertainty associated with the choice of time period. The choice of end points does not seem to induce a systematic bias, e.g. towards smaller or larger trends in any of the reanalyses, or in the difference between the reanalyses. Figures 9-10 thus confirm that the ozone-hole-induced trends are robustly captured by all four reanalysis data sets.

## 4 Discussion and Summary

Differences in the formulation of reanalysis systems and their observational inputs can lead to differences in their representation of the atmosphere, particularly for variables that are not directly observed (Fujiwara et al., 2017). Given the relatively limited observations over Antarctica, there is greater potential for spread in their representation of the Southern Hemisphere circulation response to the ozone hole. Our results suggest that there is nonetheless a high degree of consistency across the four reanalysis datasets in the representation of the dynamical changes associated with ozone depletion during 1980 to 2001. This conclusion is based on a thorough assessment of trends in the zonally averaged zonal wind, eddy heat flux, eddy momentum flux, wave forcing, and Coriolis torque (Figures 1-9). Nevertheless, trends in eddy terms that are less well constrained by available observations were found to be in close agreement for a wide range of diagnostics (Figure 10).

The expected strengthening of the lower stratospheric polar vortex during the austral spring-summer season and poleward shift of the polar front jet in the troposphere during summer in response to the ozone hole is apparent in all four reanalyses (Figures 1-3 and A2). The differences in the trends in zonal wind between ERA5, JRA-55 and MERRA-2 are generally small in both the lower stratosphere and troposphere, with the largest differences of the order 0.2 m s$^{-1}$ dec$^{-1}$, which is small compared to the size of the reanalysis ensemble mean trends (up to 5 m s$^{-1}$ dec$^{-1}$ in the stratosphere and 2 m s$^{-1}$ dec$^{-1}$ in the troposphere). CFSR, however, shows greater disagreement compared to the other three reanalyses, evident by a relatively stronger wind increase in the lower stratosphere in spring and a larger poleward displacement of the polar front jet in summer (resulting in differences in the troposphere of up to 1 m s$^{-1}$ dec$^{-1}$). These results are consistent with Dong et al. (2020), who examined near-surface summer wind speed trends for the 1980-2018 period over Antarctica in six reanalysis products (including ERA5, JRA-55, MERRA-2, and CFSR), and also found differences in the magnitude of wind speed trends.

The good agreement between ERA5 and JRA-55 circulation trends is perhaps because they both employ a 4D-VAR assimilation scheme, which is more sophisticated than the 3D-FGAT scheme employed by MERRA-2 and the 3D-VAR scheme employed by CFSR. However, examination of the time series of lower stratosphere temperatures for spring (Craig Long, personal communication) showed that CFSR was warmer than the other three reanalyses in the 1980s, which explains why its springtime temperature trends in the lower stratosphere are more negative than the others. The reason for this is that CFSR is initialized by NCEP-NCAR Reanalysis 1 (Kistler et al., 2001), which is also too warm in the 1980s in spring and the lower stratosphere (Craig Long, personal communication). Disagreements between the reanalyses could also depend on the observations that they assimilate (Manney et al., 2005; Lawrence et al., 2015). Long et al. (2017) shows that disagreements between reanalyses in the lower stratosphere temperature at Southern Hemisphere high-latitudes are greater during the period 1979 to 1998 (corresponding to the assimilation of TIROS Operational Vertical Sounder (TOVS) data), which largely corresponds to the period examined in this study, and less afterwards during the ATOVS (Advanced TOVS) period from 1999 to 2014. The ability of each reanalysis to transition seamlessly between different satellite and other data sources at different times is also an issue, with more recent reanalysis having fewer discontinuities (Long et al., 2017).

The representation of ozone and its radiative feedback also varies widely between reanalyses and might be an additional factor (Davis et al., 2017). For example, JRA-55, MERRA-2 and CFSR feed the assimilated ozone field to the radiation scheme of the reanalysis forecast model, enabling some ozone-temperature feedback (Davis et al., 2017). However, in ERA5 the ozone field fed to the radiation scheme is based on an ozone climatology, i.e., the impact of ozone depletion associated with the ozone hole on temperature is missing (Hersbach et al., 2020). The primary reason for the assimilation of ozone is that satellite sounder infrared radiances include a contribution from ozone, so knowing the ozone amount helps the radiative transfer code account for that part of the infrared spectrum and thus the thermal contribution (Craig Long, personal communication). However, the assimilated ozone data are generally not available during the long Antarctic polar night, so much of the observed depletion of stratospheric ozone in late winter associated with the ozone hole is not being properly assimilated (Davis et al., 2017).

The circulation changes are consistent with our dynamical understanding of the stratosphere-troposphere system and are explainable in terms of four stages, which are apparent in all four reanalyses. An initial strengthening of the circulation in response to radiative cooling during the 'onset' stage plays an important role in conditioning the polar vortex so that fewer planetary waves can propagate into the stratosphere from the troposphere. The strengthening of stratospheric vortex winds in spring (mainly November) during the 'growth' stage is associated with a positive trend in EPFD (Figure 4). This corresponds with reduced upward planetary wave activity fluxes at high latitudes from the troposphere into the lower stratosphere, causing a reduction in the wave-driven deceleration of the polar vortex (Figures 5 and 6). The weakening of the strengthened vortex in summer during the 'decline' and 'decay' stages is associated with a negative trend in EPFD (Figure 4). This coincides with increased upward planetary wave activity fluxes from the troposphere into the lower stratosphere at high latitudes due to the delayed breakdown of the stratospheric vortex, causing an increase in the wave-driven deceleration of the polar vortex (Figures

7 and 8). Both positive and negative trends in EPFD descend towards the tropopause, indicating a feedback between the strength of the vortex and the propagation of planetary waves (Chen and Robinson, 1992). The strengthening and poleward-displacement of the polar front jet in the troposphere during the 'decline' and 'decay' stages are robustly captured by all four reanalysis data sets and these processes are associated with changes to the synoptic-scale eddy fluxes of momentum and heat that drive the tropospheric annular modes, which is evident by enhanced poleward eddy momentum fluxes into the jet (Figures 7 and 8).

Consistent with quasi-geostrophic scaling, trends in the Coriolis torque on the residual circulation were nearly in balance with opposite trends in the eddy momentum divergence (EPFD term), as shown in Figure 4. These changes in wave forcing and wave propagation are described by Orr et al. (2012, 2013), as well as other studies such as Hartmann et al. (2000), McLandress et al. (2010, 2011), and Hu et al. (2015). They agree with the temporal evolution of the zonal wind trends, but do not indicate causality. The origin of wind anomalies begins with the slumping of angular momentum surfaces in response to changes in radiative heating by ozone, i.e., the movement of mass to maintain thermal wind balance. The total response depends further on feedback with the resolved eddy forcing, changes in parameterized gravity wave drag, and other ageostrophic terms in the momentum budget. For example, the poleward displacement and intensification of the tropospheric polar front jet in response to the ozone hole is likely to have changed Southern Hemisphere unresolved sources of orographic gravity waves generated by flow impinging on Antarctica (e.g., Hoffmann et al., 2016) and non-orographic gravity waves generated by Southern Ocean storm tracks (e.g., Charron and Manzini, 2002), resulting in changes to the momentum fluxes and drag. However, seperating the influence of gravity wave drag, the impact of reanalysis increments, and other residual terms is beyond the scope of the manuscript; as we have used a dataset interpolated to a common grid for the most consistent comparison of the reanalyses, and lack access to all the necessary terms in the residual. This should be the topic of future work. None-the-less, we emphasize the consistency of the dominant balance of the eddy terms with the zonal mean trends, despite the fact that the latter are better constrained by available observations. This internal consistency gives us greater confidence in the overall reanalysis trends.

It is found that, although the circulation trends are generally similar from one reanalysis to the next (with the exception of CFSR), important/large discrepancies in the EPFD trends in the troposphere among the four reanalyses show up as alternating negative and positive horizontal banding (Figure 4), which can be greater than the size of the mean trends across all reanalyses. Lu et al. (2015) suggest that the main contributor for such discrepancies are differences in the vertical derivative of the temperature, which are related to known issues with temperature increments caused by systematic biases in the assimilation of satellite measurements (e.g., Kobayashi et al., 2009). This is consistent with the discrepancies in temperature trends among the four reanalyses, which form vertically alternating negative and positive horizontal bands (Figure A1). An additional factor could also be that derivatives are sensitive to interpolation from model levels to standard pressure levels. However, as there are no vertically alternating differences in the tropospheric wind trend, this suggests that this potential issue is the zonal winds

are relatively well constrained by analysis increments during data assimilation, while the EPFD is more model dependent. In the lower stratosphere, the trend in EPFD shows little difference among the four reanalyses.

The disparity between the size of the differences in wind trend and differences in eddy fluxes is also apparent. There are important/large discrepancies in the associated trends in the eddy heat flux during the 'growth' stage (in November) and the 'decline' and 'decay' stages (in DJF) in the lower stratosphere, and the eddy momentum flux during the 'decline' and 'decay' stages in the troposphere (Figures 4-8). For CFSR, the positive trend in eddy heat flux during November is around half that of ERA5 (Figure 5 and 6), indicating a much weaker reduction in upward wave activity / smaller reduction in wave-driven

deceleration, despite it showing stronger positive wind trends in the lower stratosphere compared to the other reanalyses (Figure 2), which is dynamically inconsistent. This suggests that the eddy fluxes are also less constrained by the assimilation of observations, and that reanalysis temperature increments are able to cancel out differences in wave forcing, so that ultimately the impact on the large-scale circulation is small. Generally, across the four reanalyses, the largest disagreement is observed in the CFSR wave forcing / propagation trends compared to the other three reanalyses, while the best agreement is found

between ERA5 and JRA-55 (Figures 4-8).

To summarize, we show that all four modern reanalysis datasets provide a consistent estimate of the circulation changes due to the ozone hole, and that the discrepancies between the datasets are comparatively small. While our results show broad agreement on dynamical trends (eddy heat and momentum fluxes), there are non-trivial differences between reanalysis products, indicating that there is still room for improvement in our characterization of the atmosphere. Despite the consistency

across reanalyses, it is possible that changes in the observational network over time could lead to spurious trends across them all; they share the vast majority of the same input data. We have greater confidence in the trends in the circulation precisely because the changes can be explained by robust dynamical mechanisms. The reanalyses are both consistent with each other and self-consistent with our dynamical understanding of stratosphere-troposphere interactions. Looking forward, these findings will give us confidence that reanalysis datasets can be used to rigorously assess changes associated with the recovery

of stratospheric ozone (Solomon et al., 2016; Banerjee et al., 2020), which is projected to return to 1980 levels within the next few decades (Iglesias-Suarez et al., 2016).

**Appendix A: Temperature trends**

Temperature changes in the lower stratosphere are an important component of the ozone hole. To illustrate this, Figure A1 shows time-height cross-section of trends in zonally averaged temperature from September to February for ERA5, MERRA-

2, JRA-55, and CFSR. Understanding differences between reanalyses in the zonal wind trend near to the surface is also important, which is examined in Figure A2 by comparing DJF trends in the zonally averaged 850 hPa zonal wind.

**Code and data availability**

The ERA5, JRA-55, MERRA-2, and CFSR zonal-mean data set of diagnostics used in this study are available for download from the CEDA (Centre for Environmental Data Analysis) website: https://catalogue.ceda.ac.uk/uuid/b241a7f536a244749662360bd7839312 (Martineau, 2017).

**Competing interests**

The authors declare that they have no conflicts of interest.

**Financial support**

AO, HL, GM, and TB are supported by Natural Environment Research Council core funding to the British Antarctic Survey's Atmosphere, Ice and Climate team. EPG acknowledges support from the US NSF through award AGS-1852727.

**Author contribution**

AO prepared all the figures, with advice from HL. AO wrote the text, with advice from all co-authors.

**Acknowledgements**

We are grateful for advice from Tony Phillips on technical aspects of the data analysis, and Craig Long and Sean Davis on CFSR temperature trends and the treatment of ozone in reanalysis.

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

**Table 1. Key characteristics of ERA5, JRA-55, MERRA-2, and CFSR reanalysis systems. The abbreviations used are IFS (Integrated Forecast System), JMA GSM (Japanese Meteorological Agency Global Spectral Model), GEOS (Goddard Earth Observing System Model), NCEP CFS (National Center for Atmospheric Research Climate Forecast System), 3D-FGAT (Three Dimensional First Guess at Appropriate Time assimilation scheme), 4D-VAR (Four Dimensional Variational Data Assimilation), and 3D-VAR (Three Dimensional Variational Data Assimilation). In the column labelled 'Model' the year indicates the year for the version of the forecast model that was used for the reanalysis.**

| Reanalysis | Reference | Model | Horizonal grid spacing | Vertical resolution | Assimilation scheme |
|---|---|---|---|---|---|
| ERA5 | Hersbach et al. (2020) | IFS Cy41r2 (2016) | ~ 31 km | 137 levels up to 0.01 Pa | 4D-VAR (Hersbach et al., 2020) |
| JRA-55 | Kobayashi et al. (2015) | JMA GSM (2009) | ~ 55 km | 60 levels up to 0.1 hPa | 4D-VAR (Kobayashi et al., 2015) |
| MERRA-2 | Gelaro et al. (2017) | GEOS 5.12.4 (2015) | $0.5° \times 0.625°$ | 72 levels up to 0.1 hPa | 3D-FGAT (Lawless et al., 2010) |
| CFSR | Saha et al. (2010, 2014) | NCEP CFS (2007) | $0.3125°$ | 64 levels up to 0.26 hPa | 3D-VAR (Saha et al., 2010, 2014) |

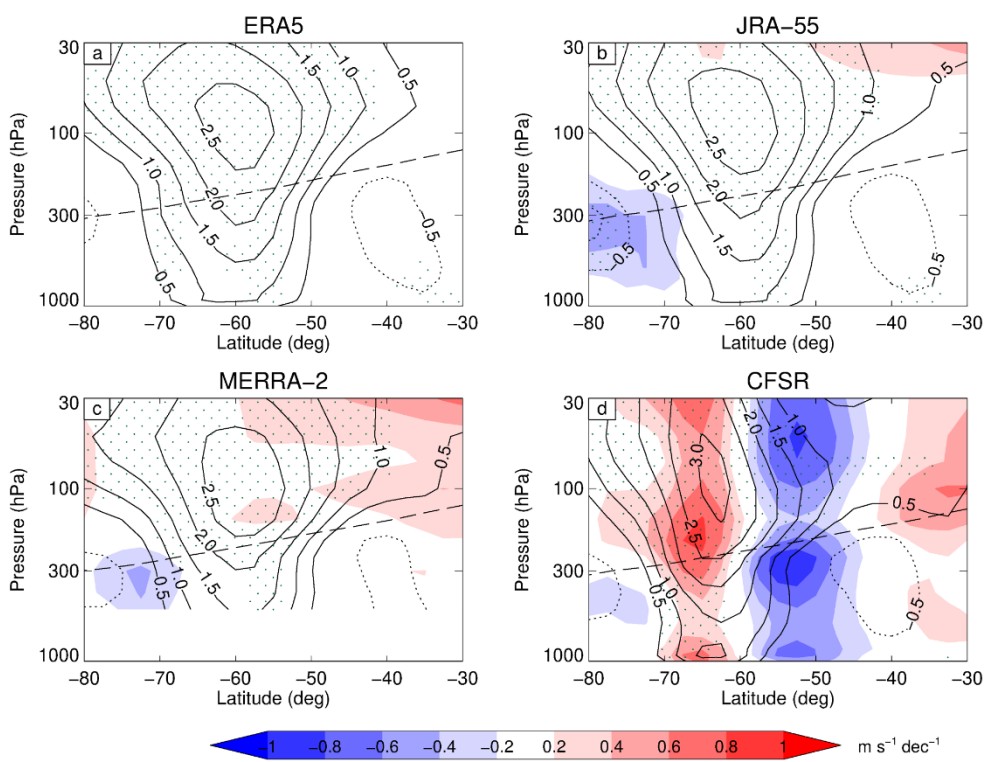

**Figure 1: DJF trend of the zonally averaged zonal wind (contour intervals: ±0.5, ±1.0, ±1.5, ±2.0, ±2.5, ±3.0 m s$^{-1}$ dec$^{-1}$) from 1980 to 2001 for ERA5 (a), JRA-55 (b), MERRA-2 (c), and CFSR (d). The shading represents differences from ERA5 at intervals of ±0.2, ±0.4, ±0.6, ±0.8, ±1.0 m s$^{-1}$ dec$^{-1}$. The dashed line shows the climatological tropopause level. Results in the range 500 to 1000 hPa are not included in panel (c). Stippling denotes regions where the trends are significant at the 5% significance level.**

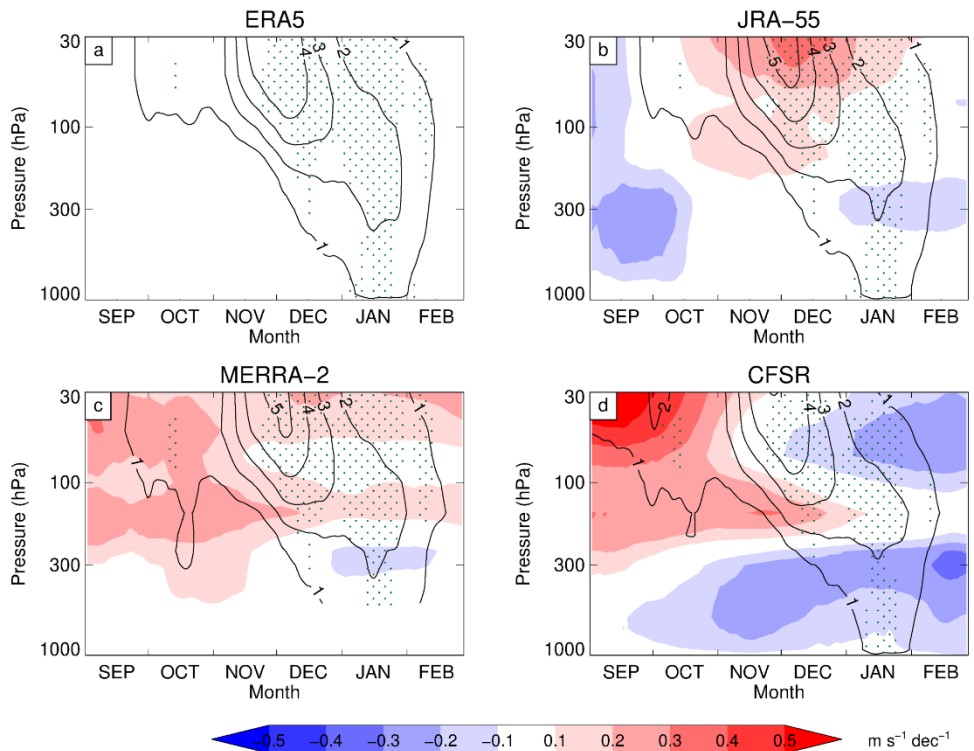

**Figure 2: Time-height cross section of the trend in the zonally averaged zonal wind (contour intervals: 1, 2, 3, 4, 5 m s⁻¹ dec⁻¹) averaged over 50 to 70°S from 1980 to 2001 for ERA5 (a), JRA-55 (b), MERRA-2 (c), and CFSR (d). The shading represents differences from ERA5 at intervals of ±0.1, ±0.2, ±0.3, ±0.4, ±0.5 m s⁻¹ dec⁻¹. Results in the range 500 to 1000 hPa are not included in panel (c). Stippling denotes regions where the trends are significant at the 5% significance level. Note that for each panel the time-series is smoothed.**

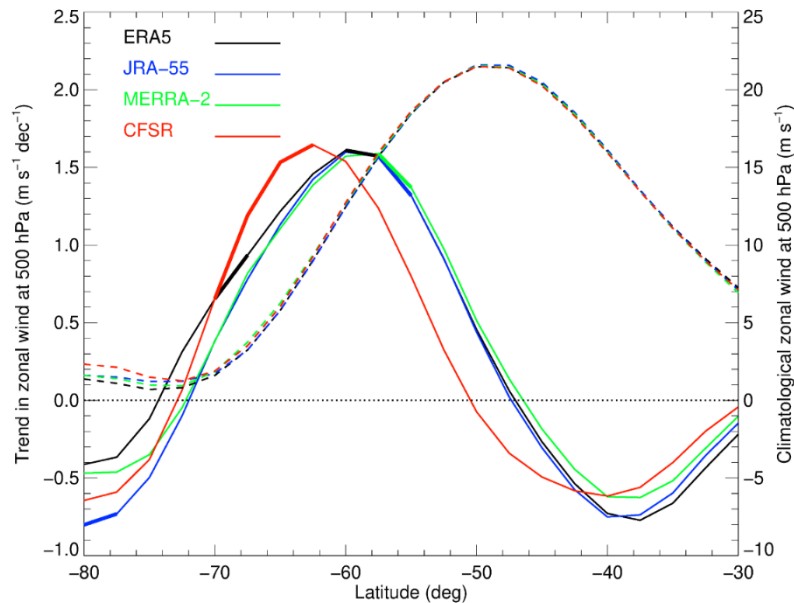

**Figure 3: DJF trend and mean in zonally averaged 500 hPa zonal wind from 1980 to 2001 for ERA5 (black line), JRA-55 (blue line), MERRA-2 (green line), and CFSR (red line). The trend is indicated by the solid lines (left y axis; units: m s⁻¹ dec⁻¹) and the climatological mean by the dashed lines (right y axis; units: m s⁻¹). Thick solid lines denote latitudes where the trends are significant at the 5% significance level. Note that the right and left axes have different scales.**

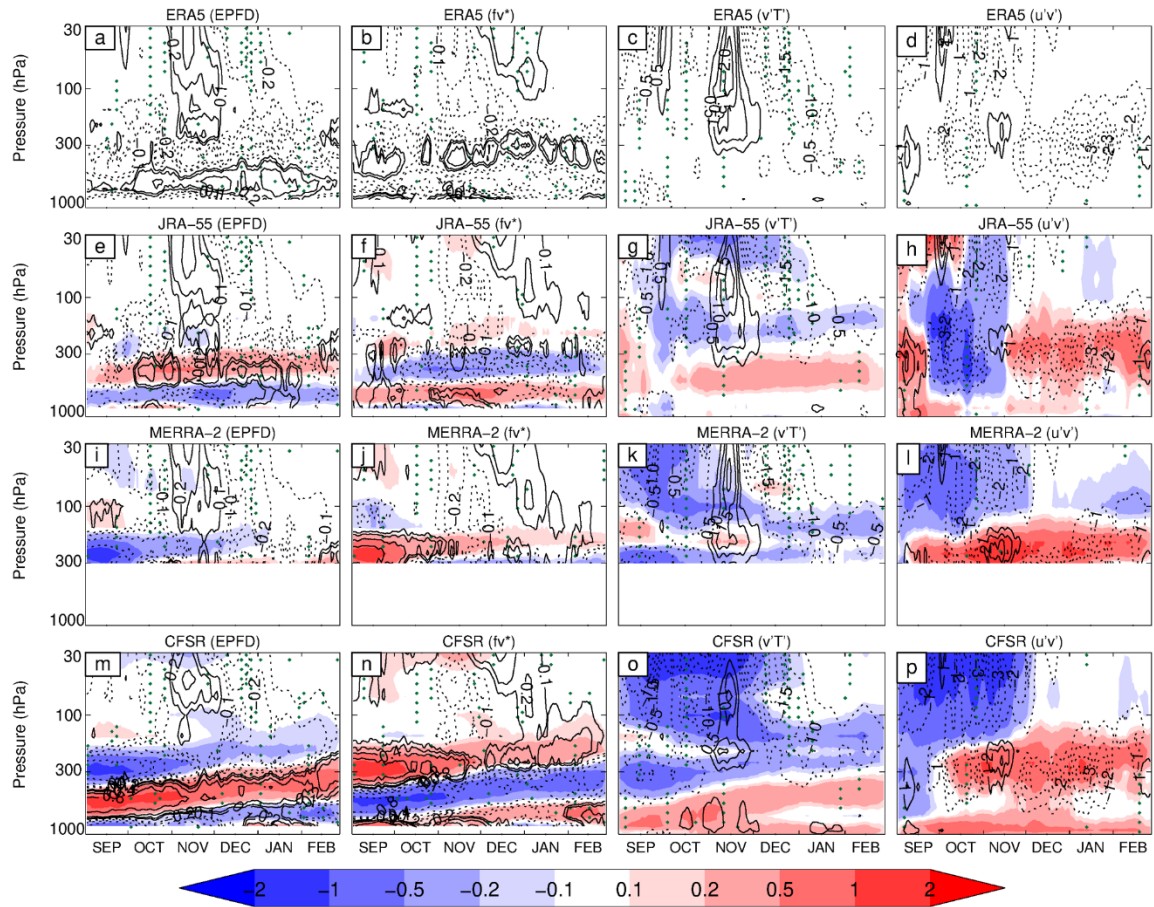

**Figure 4: Time-height cross section of the trends in EP flux divergence EPFD (contour units: ±0.1, ±0.2, ±0.4, ±0.8 m s⁻¹ d⁻¹ dec⁻¹; first column), Coriolis torque $f\bar{v}^*$ (contour units: ±0.1, ±0.2, ±0.4, ±0.8 m s⁻¹ d⁻¹ dec⁻¹; second column), eddy heat flux $\overline{v'T'}$ (contour units: ±0.5, ±1.0, ±1.5, ±2.0, ±2.5, ±3.0 m s⁻¹ K dec⁻¹; third column), and eddy momentum flux $\overline{u'v'}$ (contour units: ±1, ±2, ±3, ±4, ±5 m² s⁻² dec⁻¹; fourth column) averaged over 40 to 80°S from 1980 to 2001 for ERA5 (a, b, c, d), JRA-55 (e, f, g, h), MERRA-2 (i, j, k, l), and CFSR (m, n, o, p). The shading represents differences from ERA5 at intervals of ±0.1, ±0.2, ±0.5, ±1.0, ±2.0. Results in the range 300 to 1000 hPa are not included in panels (i), (j), (k) and (l). Stippling denotes regions where the trends are significant at the 5% significance level. Note that for each panel the time-series is smoothed.**

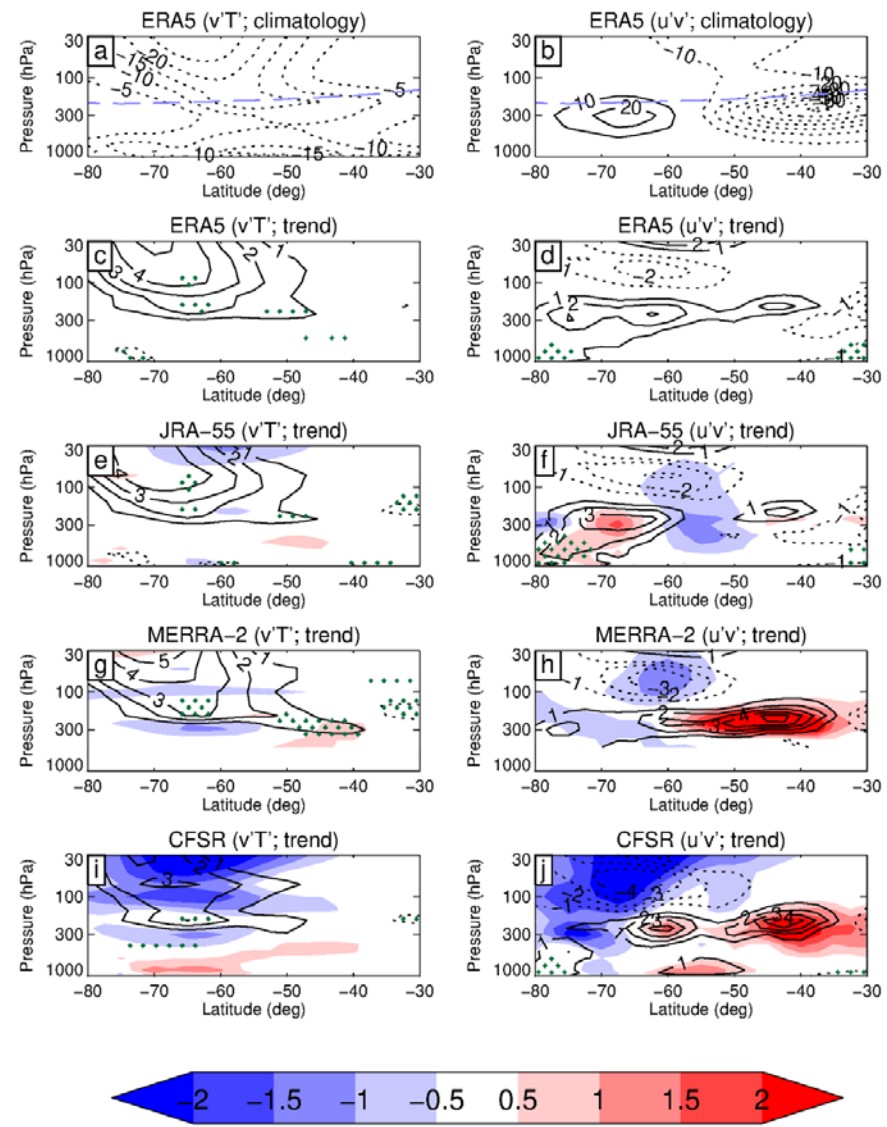

**Figure 5: November trend of eddy heat flux $\overline{v'T'}$ (contour units: ±1.0, ±2.0, ±3.0, ±4.0, ±5.0 m s$^{-1}$ K dec$^{-1}$; left column) and eddy momentum flux $\overline{u'v'}$ (contour units: ±1.0, ±2.0, ±3.0, ±4.0 m$^2$ s$^{-2}$ dec$^{-1}$; right column) from 1980 to 2001 for ERA5 (c, d), JRA-55 (e, f), MERRA-2 (g, h), and CFSR (i, j). The shading represents differences from ERA5 at intervals of ±0.5, ±1.0, ±1.5, ±2.0. Note that results in the range 500 to 1000 hPa are not included in panels (g, h). Panels (a, b)**

**show the climatological mean values of $\overline{v'T'}$ (m s$^{-1}$ K) and $\overline{u'v'}$ (m$^2$ s$^{-2}$) for ERA5 from 1980 to 2001, with the blue dashed line indicating the climatological tropopause level. Stippling denotes regions where the trends are significant at the 5% significance level**

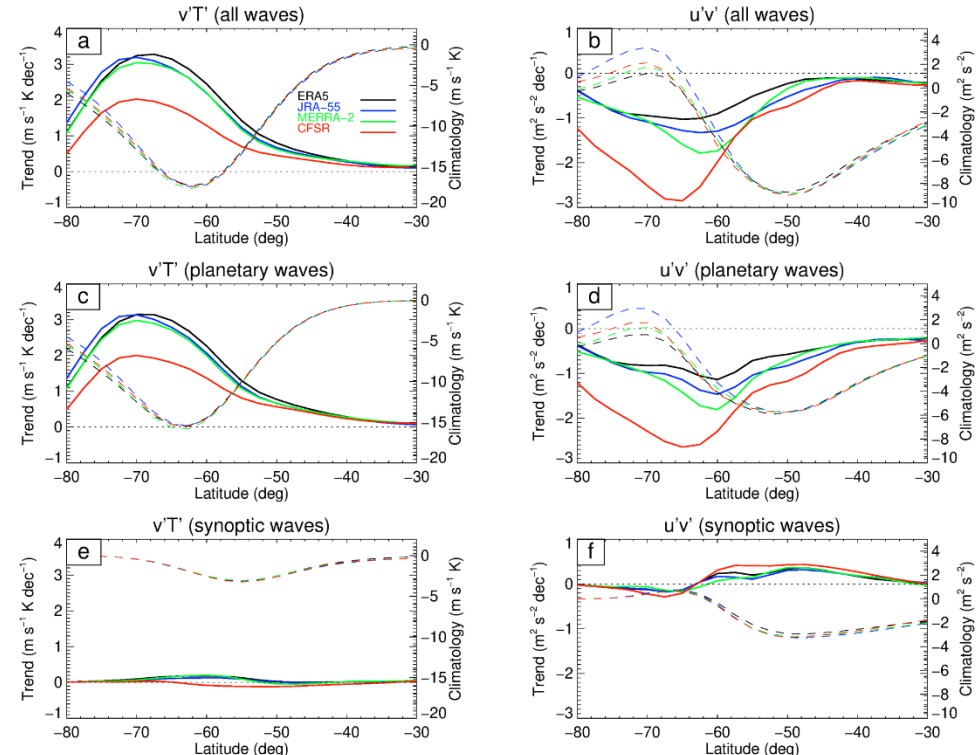

**Figure 6:** November trend and climatological mean in 30 to 100 hPa vertically integrated eddy heat flux $\overline{v'T'}$ (left column) and eddy momentum flux $\overline{u'v'}$ (right column) due to all waves (a, b), planetary waves (c, d) and synoptic waves (e, f) from 1980 to 2001 for ERA5 (black), JRA-55 (blue), MERRA-2 (green), and CFSR (red). The trend in $\overline{v'T'}$ is indicated by the thick lines (left y axis; units: m s$^{-1}$ K dec$^{-1}$) and the climatological mean by the thin dashed lines (right y axis; units: m s$^{-1}$ K). The trend in $\overline{u'v'}$ is indicated by the thick lines (left y axis; units: m$^2$ s$^{-2}$ dec$^{-1}$) and the climatological mean by the thin dashed lines (right y axis; units: m$^2$ s$^{-2}$). None of the trends are significant at the 5% significance level. Note that for both columns the right and left axes have different scales.

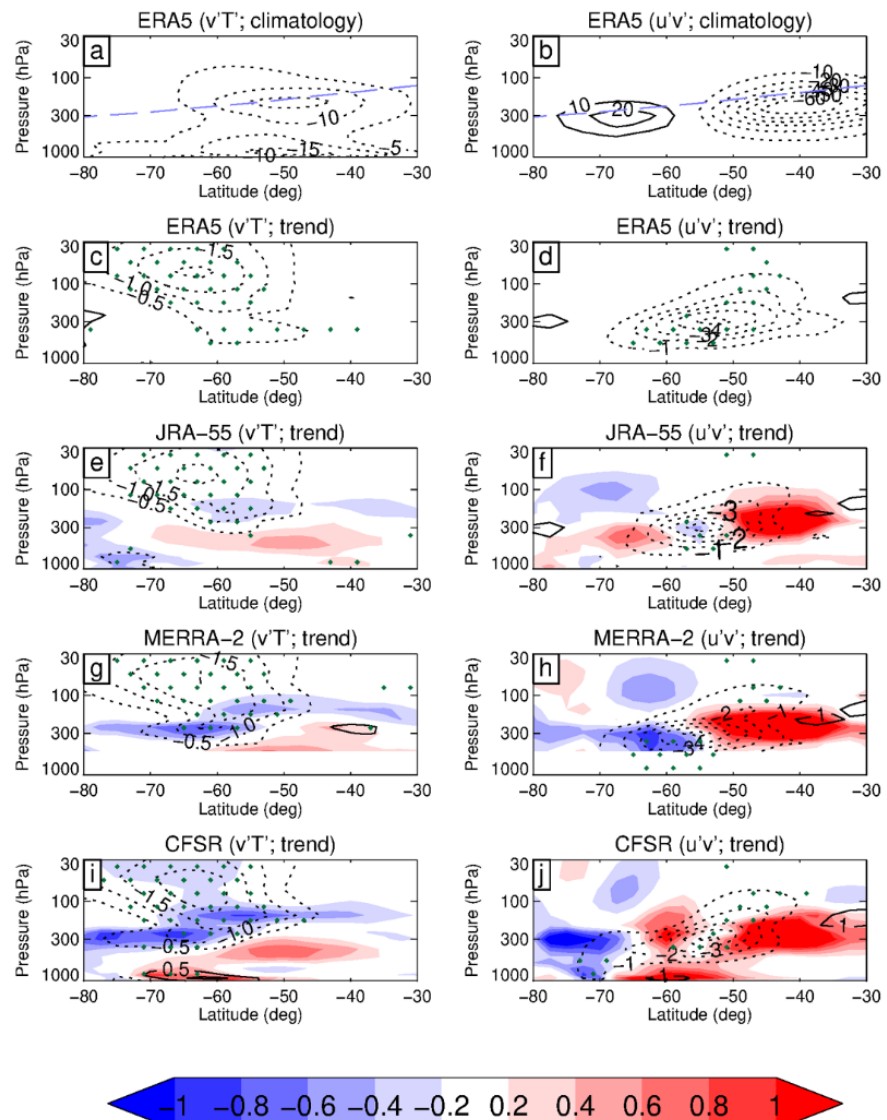

**Figure 7: DJF trend of eddy heat flux $\overline{v'T'}$ (contour units: ±0.5, ±1.0, ±1.5, ±2.0 m s⁻¹ K dec⁻¹; left column) and eddy momentum $\overline{u'v'}$ flux (contour units: ±1.0, ±2.0, ±3.0, ±4.0, ±5.0, ±6.0 m² s⁻² dec⁻¹; right column) from 1980 to 2001 for ERA5 (c, d), JRA-55 (e, f), MERRA-2 (g, h), and CFSR (i, j). The shading represents differences from ERA5 at intervals of ±0.2, ±0.4, ±0.6, ±0.8, ±1.0. Note that results in the range 500 to 1000 hPa are not included in panels (g, h). Panels (a, b) show the climatological mean values of $\overline{v'T'}$ (m s⁻¹ K) and $\overline{u'v'}$ (m² s⁻²) for ERA5 from 1980 to 2001, with the blue dashed line indicating the climatological tropopause level. Stippling denotes regions where the trends are significant at the 5% significance level.**

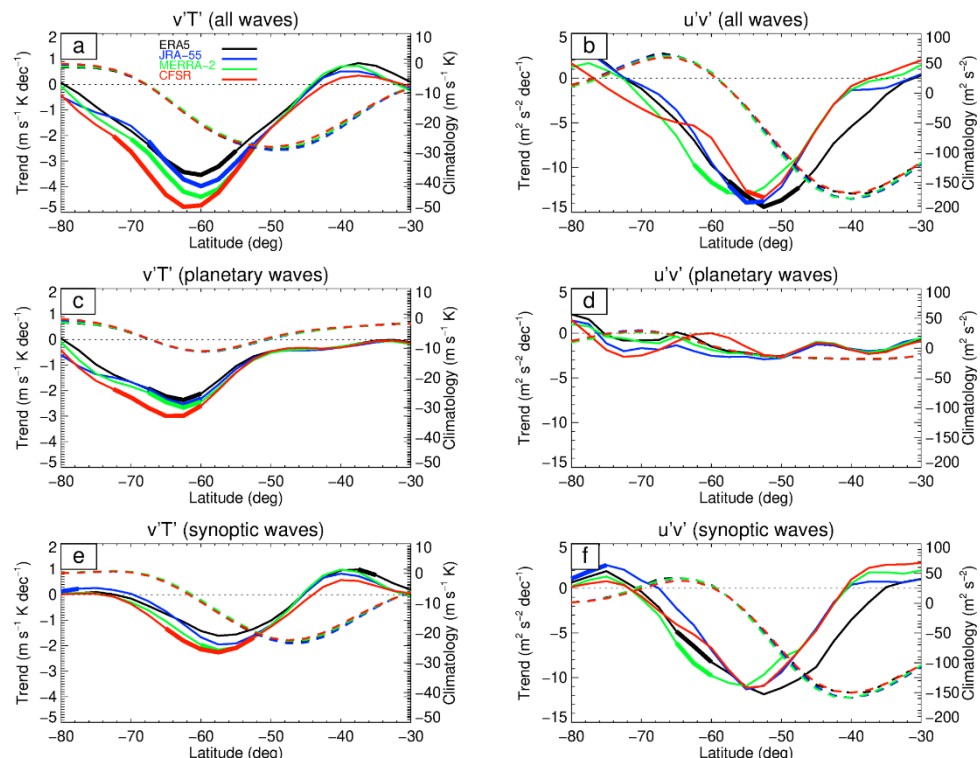

**Figure 8: DJF trend and climatological mean in vertically integrated zonally eddy heat flux $\overline{v'T'}$ from 30 to 300 hPa (left column) and eddy momentum flux $\overline{u'v'}$ from 100 to 500 hPa (right column) due to all waves (a, b), planetary waves (c, d) and synoptic waves (e, f) from 1980 to 2001 for ERA5 (black), JRA-55 (blue), MERRA-2 (green), and CFSR (red). The trend in $\overline{v'T'}$ is indicated by the thick lines (left y axis; units: m s$^{-1}$ K dec$^{-1}$) and the climatological mean by the thin dashed lines (right y axis; units: m s$^{-1}$ K). The trend in $\overline{u'v'}$ is indicated by the thick lines (left y axis; units: m$^2$ s$^{-2}$ dec$^{-1}$) and the climatological mean by the thin dashed lines (right y axis; units: m$^2$ s$^{-2}$). Thick solid lines denote latitudes where the trends are significant at the 5% significance level. Note that for both columns the right and left axes have different scales.**

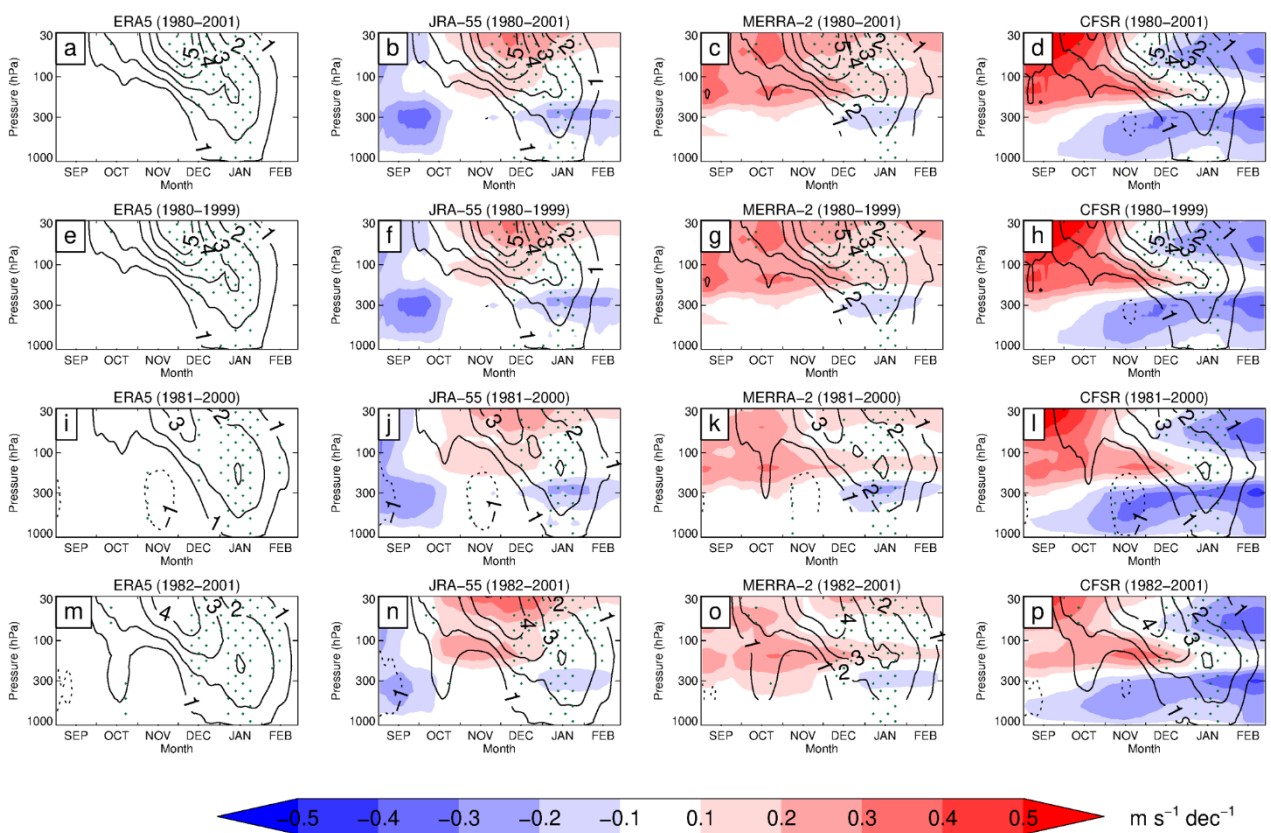

**Figure 9: Sensitivity of the trend in zonal wind to time period, displayed as time-height cross section of the trend in the zonally averaged zonal wind (contour intervals: 1, 2, 3, 4, 5 m s⁻¹ dec⁻¹) averaged over 50 to 70°S from 1980 to 2001 (a-d; same results as shown in Figure 2), 1980 to 1999 (e-h), 1981 to 2000 (i-l) and 1982 to 2001 (m-p) for ERA5, JRA-55, MERRA-2, and CFSR. The shading represents differences from ERA5 at intervals of ±0.1, ±0.2, ±0.3, ±0.4, ±0.5 m s⁻¹ dec⁻¹. Results in the range 500 to 1000 hPa are not included in panels (c, g, k, o). Panels (a-d) are the same results as shown in Figure 2. Stippling denotes regions where the trends are significant at the 5% significance level. Note that for each panel the time-series is smoothed.**

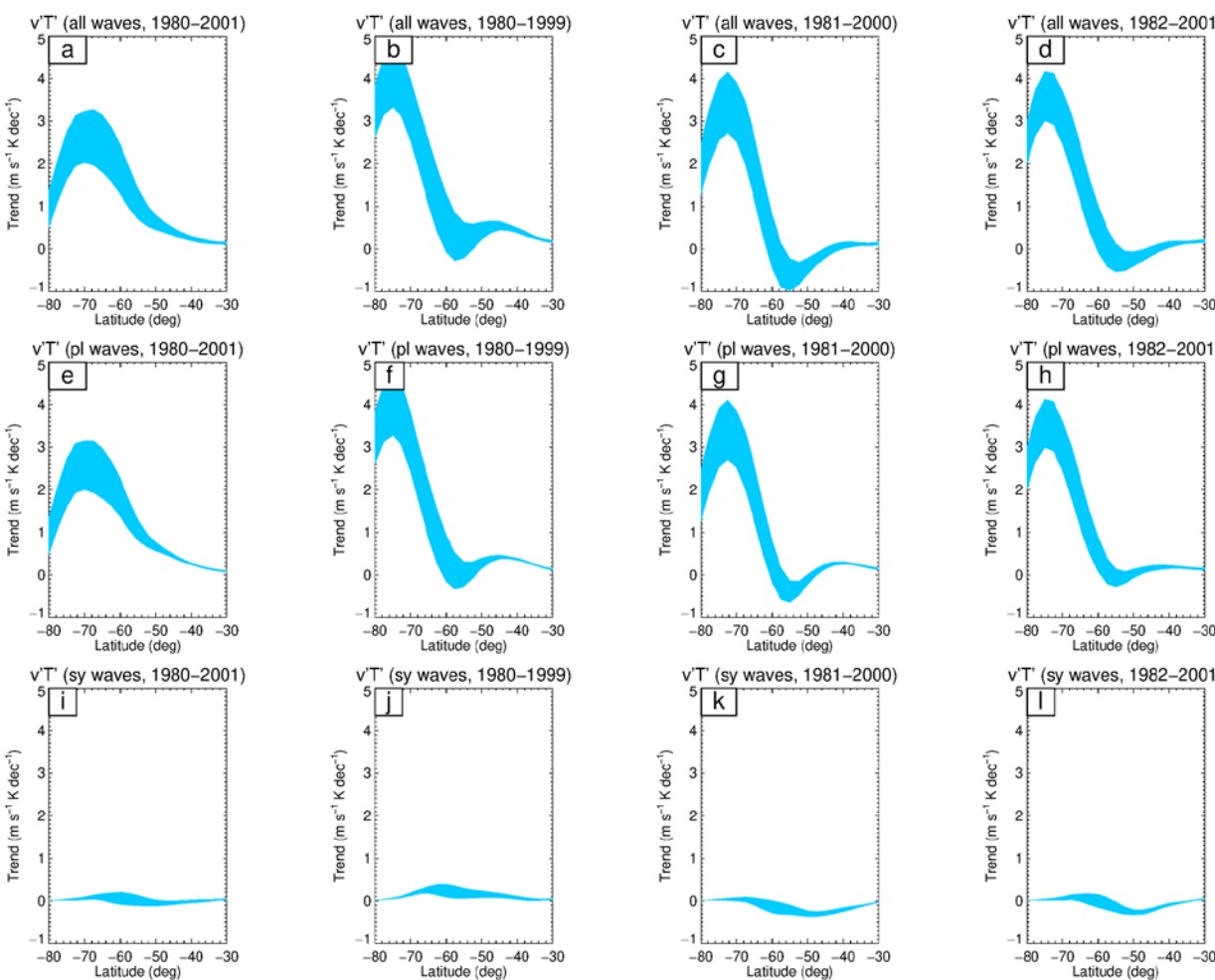

**Figure 10:** Sensitivity of the trends in eddy heat flux to time period, displayed as shaded envelopes representing the spread (maximum and minimum values) derived from ERA5, JRA-55, MERRA-2 and CFSR of the November trend in 30 to 100 hPa vertically integrated eddy heat flux $\overline{v'T'}$ from 1980 to 2001 (a, e, i), 1980 to 1999 (b, f, j), 1981-2000 (c, g, k) and 1982-2001 (d, h, i) due to all waves (top row), planetary waves (middle row), and synoptic waves (bottom row). The units are m$^2$ s$^{-2}$ dec$^{-1}$. Panels (a, e, i) are the same results as shown in the left column of Figure 6.

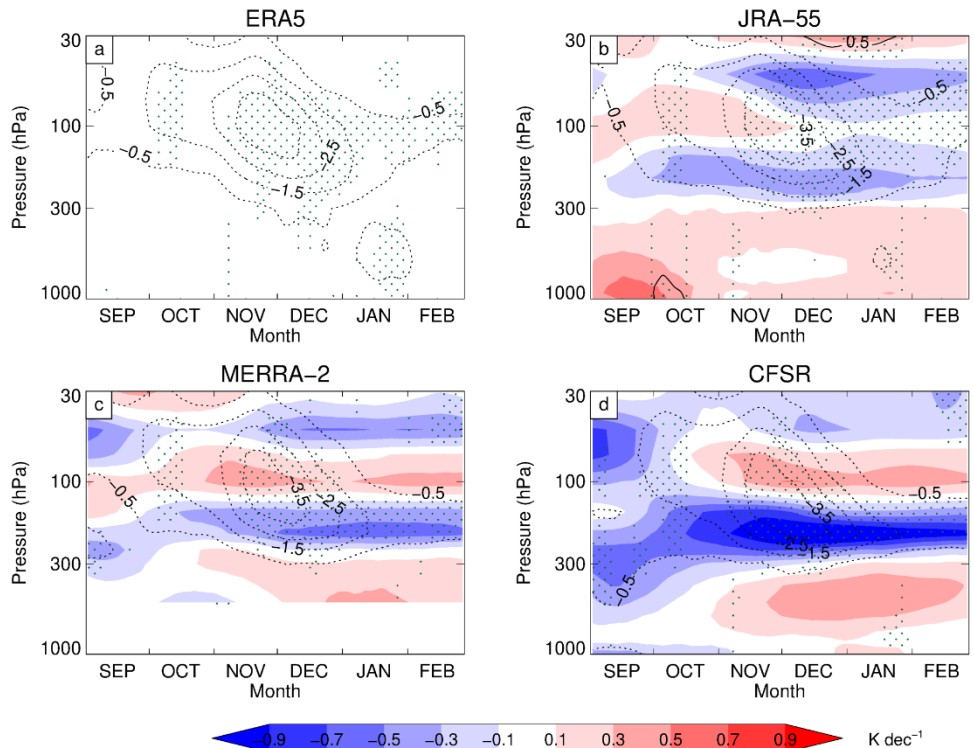

**Figure A1: Time-height cross section of the trend in the zonally averaged temperature (contour intervals: -0.5, -1.5, -2.5, -3.5 K dec⁻¹) averaged over 70 to 87.5°S from 1980 to 2001 for ERA5 (a), JRA-55 (b), MERRA-2 (c), and CFSR (d). The shading represents differences from ERA5 at intervals of ±0.1, ±0.3, ±0.5, ±0.7, ±0.9 K dec⁻¹. Results in the range 1000 to 500 hPa are not included in panel (c). Stippling denotes regions where the trends are significant at the 5% significance level. Note that for each panel the time-series is smoothed.**

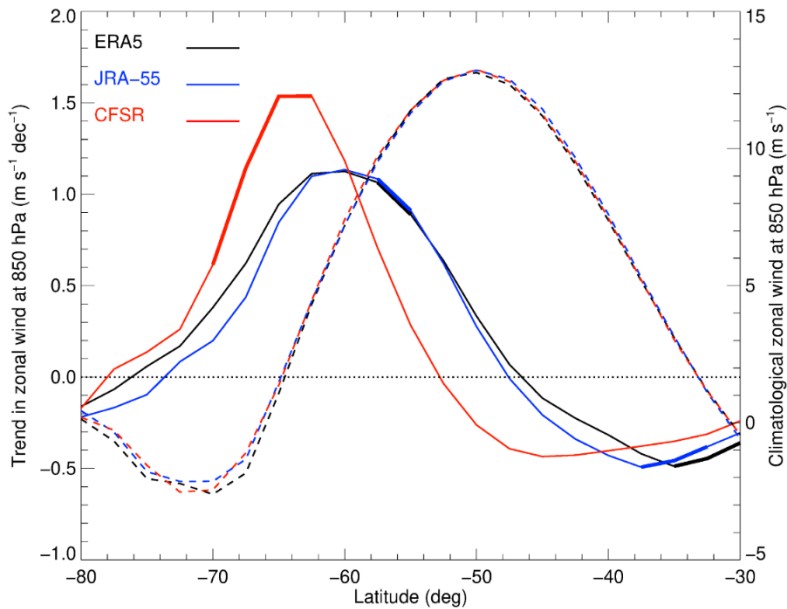

**Figure A2: As Figure 3, but showing the DJF trend and mean in zonally averaged 850 hPa zonal wind from 1980 to 2001 for ERA5 (black line), JRA-55 (blue line), and CFSR (red line). Thick solid lines denote latitudes where the trends are significant at the the 5% significance level. Note that results for MERRA-2 at this pressure level were not availble.**
