# Peer review of "Is our dynamical understanding of the circulation changes associated with the Antarctic ozone hole sensitive to the choice of reanalysis dataset?"

_Atmospheric Chemistry and Physics, 2020_

## Referee Comment (RC1) · Anonymous Referee #1 · 23 Jan 2021

Review of "Is our dynamical understanding of the circulation changes associated with the Antarctic ozone hole sensitive to the choice of reanalysis dataset?"

In this study the authors compare the response of the Antarctic polar vortex to ozone depletion over the years 1980-2001 among four current reanalysis products. The study is well motivated, very well written and, for the most part, most of the main conclusions are supported by the text and figures. To this end, I recommend publication pending minor revisions. At the same time, however, the analysis does omit certain points (i.e. exclusion of parameterized waves in wind budgets) that should be addressed more

explicitly both in the text and through incorporation of a new supplementary figure (see comments below).

Major Comment: The budget analysis does not include the effect of parameterized gravity wave drag. Nor is the tendency due to the potential imbalance caused by the incremental analysis during data assimilation included. While neglecting these terms may not matter much in the lower stratosphere, I have am not convinced that neglecting these terms is likewise trivial in the upper/middle polar stratosphere. At present, though, because the residuals are not shown the reader has no way of evaluating how well the zonal wind budgets are closed in the upper/middle stratosphere. In particular, is it possible that one of the main drivers of the differences between CSFR and the other reanalysis datasets is the contribution from parameterized GWD? Have the authors done the analysis? Is there really no way of getting access to these missing tendencies? If the answer to the latter is no, then at the very least the authors should provide a new supplementary figure that shows how well they can balance the budget for each reanalysis product.

Minor Comments:

1. Line 109: It is a bit odd that there is no mention of vertical resolution in the text nor in Table 1, especially since one would expect this to have an important impact on the representation of the (wave-driven) residual mean circulation. How does vertical resolution differ among the reanalyses? At the very least this information should be added as a new column in the table.

2. Missing primes in labels in Figures 4-8.

3. Line 133-134: The authors write that they do not examine the individual EP flux components as they "requir[e] the vertical derivative of temperature. . .resulting in noisy wave driving". And yet, later on they examine the EP flux divergence (which will partly reflect some of this undesirable noisiness, albeit somewhat smoothed). The analysis of the EP flux divergence is certainly fine but I would suggest that the authors rephrase
their earlier caveat because it seems inconsistent with the EP flux divergence analysis presented later on in the manuscript.

4. The dynamical inconsistency exhibited by CFSR (i.e. weaker reduction in upward wave activity despite stronger positive wind trend) is an important result as it exemplifies why caution is needed when analyzing momentum budgets in systems utilizing data assimilation. However, assimilation issues aside, how much of this apparent inconsistency just reflects a lack of correctly accounting for the tendency contributed by (unresolved) gravity waves? I suppose the authors address this in Line 417 but the description is extremely brief and speculative. Is there really no way to access these terms (see Major Comment)?

5. Line 420: In reference to the sentence beginning with "They showed that the sum of the wave driving...". Is this true throughout the vertical? Is the contribution of parameterized waves really not important in the middle and upper stratosphere?

6. The authors never explicitly show what the differences in the trends in the polar vortex and upper troposphere/lower stratosphere imply for the surface trends. Is the SH surface jet response also anomalous for CFSR? This seems like an important point that should be discussed.

---

## Referee Comment (RC2) · Anonymous Referee #2 · 25 Jan 2021

The study examines differences among four reanalyses in the representation of the circulation response to the ozone hole in Austral spring and summer. The results show overall good consistency in the circulation trends, slightly worse in the Eliassen-Palm flux divergence and eddy heat and momentum fluxes. The CFSR reanalysis is shown to diverge most from the other three reanalyses.

The paper is well written and results are consistent with previous works, and a valuable contribution to S-RIP activity. Nevertheless, I consider that the following comments should be addressed before publication.

*General comments:*

- The paper would be more complete if all terms in the momentum balance (Eq. 1) were evaluated. As mentioned in the discussion with reference to Orr et al. 2012, the balance is mainly between the Coriolis torque and the wave drag terms. Therefore, including these additional terms would provide interesting information for the S-RIP activity regarding the representation of the residual circulation trends in response to the ozone hole across reanalyses, as well as non-resolved waves and the role of assimilation increments (included in the residual).

- In the Data Section, it would be helpful to discuss which reanalyses assimilate ozone observations into the model, that is, if the radiative code 'sees' the ozone hole and therefore the described feedbacks are captured in the model, or if in contrast the response is artificially forced by the assimilation increments. This is only briefly mentioned in the discussion (L377-378). Based on your results, is this an important factor?

- Statistical significance of the trends is not discussed anywhere in the paper. A statistical analysis should be added, not only about the trend significance, but also on the significance of the differences among reanalyses.

*Specific comments:*

- L27: what is the justification for using the quasi-geostrophic version of the EP flux?

- 133-134: It is fine to show only the fluxes, since they contain the information that you want to discuss. However I find the argument given here not satisfactory, because theta(p) is likely changing substantially due to the lower stratospheric cooling, so it could be important for the divergence (and I assume the full term is used to calculate it).

- L235: alternating stripes are also seen in the temperature trends (Fig. A1), although at different levels. I wonder if these indicate a more fundamental issue with vertical model levels or assimilation. Are these features described in previous S-RIP papers? -

L264-265: "persistent negative values from December to February," except in November at tropopause levels, as you describe briefly in L300-301. Since this is a robust feature, I believe it should be discussed a bit more. Note that there is some spread among reanalyses in the location and strength of these positive momentum flux trends.

- L403-404: but there are stripes in temperature trends (see comment on L235)

*Technical corrections:*

- CFSR is misspelled (CSFR) in several occasions throughout the paper

- L164: easterly winds → easterly wind trends

- L242: overbar missing over v'T'

- L261: add (i.e. poleward momentum transfer) after SH to avoid confusion when comparing to your results.

- L270: upward → upward-propagating?

-L272: fluxes → flux

- L335: exhibit missing an 'i'

- L398: "It is found that," (add comma)

---

## Author Comment (AC1) · 6 Apr 2021

*Review of "Is our dynamical understanding of the circulation changes associated with the Antarctic ozone hole sensitive to the choice of reanalysis dataset?"*

**Anonymous Referee #1**

*In this study the authors compare the response of the Antarctic polar vortex to ozone depletion over the years 1980-2001 among four current reanalysis products. The study is well motivated, very well written and, for the most part, most of the main conclusions are supported by the text and figures. To this end, I recommend publication pending minor revisions. At the same time, however, the analysis does omit certain points (i.e., exclusion of parameterized waves in wind budgets) that should be addressed more explicitly both in the text and through incorporation of a new supplementary figure (see comments below).*

**We are grateful for the time taken by the Referee to provide such well-considered and insightful comments, especially their suggestion to consider other terms of the momentum budget (which was also suggested by Referee #2). We are pleased that they found the results meaningful and the paper clearly written. We have implemented all of their comments, which are explained below and have undoubtedly strengthened the revised manuscript. We very much hope that the referee is satisfied by these improvements.**

*(1) Major Comment: The budget analysis does not include the effect of parameterized gravity wave drag. Nor is the tendency due to the potential imbalance caused by the incremental analysis during data assimilation included. While neglecting these terms may not matter much in the lower stratosphere, I am not convinced that neglecting these terms is likewise trivial in the upper/middle polar stratosphere. At present, though, because the residuals are not shown the reader has no way of evaluating how well the zonal wind budgets are closed in the upper/middle stratosphere. In particular, is it possible that one of the main drivers of the differences between CSFR and the other reanalysis datasets is the contribution from parameterized GWD? Have the authors done the analysis? Is there really no way of getting access to these missing tendencies? If the answer to the latter is no, then at the very least the authors should provide a new supplementary figure that shows how well they can balance the budget for each reanalysis product.*

**Reply: The decision to omit examination of the Residual term (e.g., made up from parameterised gravity wave drag, analysis increments, ageostrophic terms) is based on two arguments. Firstly, the use of the quasi-geostrophic TEM approximation for the momentum budget precludes us from a meaningful analysis of the residual terms in the momentum budget, i.e., it is not the most appropriate framework for analysis at that level. Secondly, we do not have access to all the necessary terms in the budget across all reanalyses to examine this, i.e., the impact of analysis increments and gravity wave drag are included in a single term, the residual.**

**The TEM framework is ideal for understanding the momentum budget, identifying the dominate balance between the Coriolis torque on the net poleward transport of mass (quantified by the residual circulation) and the transport of momentum by Rossby waves – which is the main focus of our study, and therefore we have employed the correct methodology/tool. By contrast, the Eulerian mean momentum budget is more appropriate for a detailed analysis of the residual terms, as ageostrophic terms of the same order aren't explicitly taken into account here. But the Eulerian budget obscures the dominant balance between Coriolis forcing acting on the Lagrangian mean circulation and the eddy forcing, and so would be unsuitable for our study. For example, the Coriolis torque on the Eulerian mean circulation is of opposite sign to the actual transport of mass across much of the stratosphere, as with the Ferrell cell in the troposphere. As mentioned above, we do agree that parameterized gravity wave momentum drag can be incorporated in the TEM, but we did not have complete access to the necessary data. However, separating the role of gravity wave drag and analysis increment should be the topic of future work, and is something that I am interested in pursuing.**

**We have modified the manuscript at several points to be clear to the reader about this limitation in our analysis. First, the discussion of the TEM momentum budget (Eq. 1) has been updated to more explicitly list missing terms, and we explain in more detail how the eddy terms feed into the budget at the end of section 3.1. Second, we've included new panels in Figure 4 to show that the dominant balance in the momentum equation is between the Coriolis torque and the wave-driven forcing. Finally, we've added a new paragraph in the Discussion and Summary section to emphasize that our analysis of the budget does**

not allow us to identify the cause of the anomalies. The "first cause" is the change in radiative heating, which causes angular momentum surface to slump as to remain in thermal wind balance. Our main message is that the consistency between the representation of the mean trends and the eddy terms strengthens our confidence in the overall representation of the response.

For example, some of the changes include:

- In the Data and Methods section, the following new paragraph: '*The key variables examined in Eq. (1) are the trends in the wave forcing (EPFD), and the Coriolis torque $f\overline{v}^*$. Our use of the QG TEM approximation for the momentum budget and the lack of complete access to all the relevant data, preclude us from a meaningful analysis of the trends in the residual term in the momentum budget, so therefore this term is not considered. The residual term includes both parameterized gravity wave drag (e.g., Lott and Miller, 1997; Orr et al., 2010) and reanalysis increments in the momentum budget, but also ageostrophic terms and any numerical biases in the model (which therefore cannot be separated as they are included in a single term). The TEM framework is ideal as a diagnostic tool for identifying the dominant balance between the Coriolis torque on the net poleward transport of mass (quantified by the residual circulation) and the transport of momentum by Rossby waves (quantified by the EPFD term), i.e., examining how changes in these two terms relate to changes in the zonal mean wind, which is therefore the focus of this work. On seasonal time-scales, the EPFD and Coriolis torque terms are the leading order balance in the system: momentum transfer in the free atmosphere is controlled dynamically via eddy heat and momentum transfer (Palmer, 1981).*'

- In the Discussion and Conclusions section, extensive modification to the following paragraph: '*Consistent with quasi-geostrophic scaling, trends in the Coriolis torque on the residual circulation were nearly in balance with opposite trends in the eddy momentum divergence (EPFD term), as shown in Figure 4. These changes in wave forcing and wave propagation are described by Orr et al. (2012, 2013), as well as other studies such as Hartmann et al. (2000), McLandress et al. (2010, 2011), and Hu et al. (2015). They agree with the temporal evolution of the zonal wind trends, but do not indicate causality. The origin of wind anomalies begins with the slumping of angular momentum surfaces in response to changes in radiative heating by ozone, i.e., the movement of mass to maintain thermal wind balance. The total response depends further on feedback with the resolved eddy forcing, changes in parameterized gravity wave drag, and other ageostrophic terms in the momentum budget. For example, the poleward displacement and intensification of the tropospheric polar front jet in response to the ozone hole is likely to have changed Southern Hemisphere unresolved sources of orographic gravity waves generated by flow impinging on Antarctica (e.g., Hoffmann et al., 2016) and non-orographic gravity waves generated by Southern Ocean storm tracks (e.g., Charron and Manzini, 2002), resulting in changes to the momentum fluxes and drag. However, separating the influence of gravity wave drag, the impact of reanalysis increments, and other residual terms is beyond the scope of the manuscript; as we have used a dataset interpolated to a common grid for the most consistent comparison of the reanalyses, and lack access to all the necessary terms in the residual. This should be the topic of future work. None-the-less, we emphasize the consistency of the dominant balance of the eddy terms with the zonal mean trends, despite the fact that the latter are better constrained by available observations. This internal consistency gives us greater confidence in the overall reanalysis trends.*'

Minor Comments:

*(2) Line 109: It is a bit odd that there is no mention of vertical resolution in the text nor in Table 1, especially since one would expect this to have an important impact on the representation of the (wave-driven) residual mean circulation. How does vertical resolution differ among the reanalyses? At the very least this information should be added as a new column in the table.*

Reply: The vertical information has been added as a new column in Table 1. Additionally, the following sentence has been added to Section 2 (Data and Methods) of the revised manuscript to mention the differences in vertical resolution in the main text: '*The vertical resolution of JRA-55, MERRA-2 and CSFR is broadly similar with approximately 60-70 levels from the surface up to around 0.1 hPa, whereas ERA5 uses 137 levels from the surface up to 0.01 hPa.*'

*(3) Missing primes in labels in Figures 4-8.*

**Reply: Primes have been added to Figures 4-8, and 10.**

*(4) Line 133-134: The authors write that they do not examine the individual EP flux components as they "require the vertical derivative of temperature...resulting in noisy wave driving". And yet, later on they examine the EP flux divergence (which will partly reflect some of this undesirable noisiness, albeit somewhat smoothed). The analysis of the EP flux divergence is certainly fine but I would suggest that the authors rephrase their earlier caveat because it seems inconsistent with the EP flux divergence analysis presented later on in the manuscript.*

**Reply: This is a good point. Given that there is a clear explanation in the manuscript as to why we examine the eddy momentum flux $\overline{u'v'}$, the eddy heat flux $\overline{v'T'}$, and the EP flux divergence, a further explanation as to why we do not examine the individual EP flux components is not necessary. This sentence (explaining why we do not examine the individual EP flux components) has therefore been deleted.**

*(5) The dynamical inconsistency exhibited by CFSR (i.e. weaker reduction in upward wave activity despite stronger positive wind trend) is an important result as it exemplifies why caution is needed when analyzing momentum budgets in systems utilizing data assimilation. However, assimilation issues aside, how much of this apparent inconsistency just reflects a lack of correctly accounting for the tendency contributed by (unresolved) gravity waves? I suppose the authors address this in Line 417 but the description is extremely brief and speculative. Is there really no way to access these terms (see Major Comment)?*

**Reply: In our response to your major comment, we acknowledge the limitation of this analysis based on using the quasi-geostrophic TEM framework. We suspect that that reanalysis increments or changes in the representation of ozone are the key difference here, given the inconsistency between the response of the eddies to the mean winds. (There could be some compensation between changes in parameterized gravity waves and the resolved circulation, but we didn't have ready access to the data.) Trends are extremely sensitive to changes in the observing system, and we suspect that biases in the 1980s were reduced in later decades. As we now explain in the Discussion section, a limitation of our analysis is that both gravity wave drag and the impact of reanalyses increments are included in a single term, the Residual, and that separating the contribution from these terms should be the topic of future work.**

*(6) Line 420: In reference to the sentence beginning with "They showed that the sum of the wave driving...". Is this true throughout the vertical? Is the contribution of parameterized waves really not important in the middle and upper stratosphere?*

**Reply: The dominant balance was consistently between the Coriolis torque on the residual circulation and the resolved eddy forcing. See revised Figure 4. We believe this is largely because we have restricted our analysis to below 30 hPa and to the middle to high latitudes. We expect that gravity wave torques become order one at higher levels, and potentially in the tropical atmosphere at these heights, though we have not looked explicitly. As we acknowledge in the paper, a limitation of our analysis is that both gravity wave drag and the impact of reanalyses increments are included in a single term, the Residual, and therefore we are unable to examine their separate contributions. This should be the topic of future work.**

*(7) The authors never explicitly show what the differences in the trends in the polar vortex and upper troposphere/lower stratosphere imply for the surface trends. Is the SH surface jet response also anomalous for CFSR? This seems like an important point that should be discussed.*

**Reply: This is a good point. We have added an additional figure (labelled Figure A2 in the revised manuscript) that shows the trends in the zonally averaged zonal wind at 850 hPa, i.e., analogous to Figure 3, which shows results at 500 hPa. This clearly shows that the CFSR results are still anomalous at near-surface level, which is consistent with the tropospheric response being largely barotropic. This new figure**

is explained after Figure 3 in the Results section, using the following text: *'Note that the anomalous CFSR trend in the polar front jet compared to the other reanalyses is even more apparent at 850 hPa (Figure A2), i.e., near the surface and consistent with fairly uniform (barotropic) wind trend anomalies throughout the troposphere.'*

Additionally, we have added the following new text to the Discussion section: **'***These results are consistent with Dong et al. (2020), who examined near-surface summer wind speed trends for the 1980-2018 period over Antarctica in six reanalysis products (including ERA5, JRA-55, MERRA-2, and CFSR), and also found differences in the magnitude of wind speed trends.*'

Dong, X., Wang, Y., Hou, S., Ding, M., Yin, B., and Zhang, Y.: Robustness of the recent global atmospheric reanalyses for Antarctic near-surface wind speed climatology, J. Clim., 33, 4027-4043, https://doi.org/10.1175/JCLI-D-19-0648.1, 2020.

---

## Author Comment (AC2) · 6 Apr 2021

*Review of "Is our dynamical understanding of the circulation changes associated with the Antarctic ozone hole sensitive to the choice of reanalysis dataset?"*

**Anonymous Referee #2**

*The study examines differences among four reanalyses in the representation of the circulation response to the ozone hole in Austral spring and summer. The results show overall good consistency in the circulation trends, slightly worse in the Eliassen-Palm flux divergence and eddy heat and momentum fluxes. The CFSR reanalysis is shown to diverge most from the other three reanalyses.*

*The paper is well written and results are consistent with previous works, and a valuable contribution to S-RIP activity. Nevertheless, I consider that the following comments should be addressed before publication.*

**We are pleased that the Referee found the paper to be well written and a valuable contribution to S-RIP activity. We are grateful for their insightful comments, especially their suggestion to consider other terms of the momentum budget and add statistical significance, which have undoubtedly strengthened the revised manuscript. Their comments are addressed below. We very much hope that the referee is satisfied by these improvements.**

General comments:

*(1) The paper would be more complete if all terms in the momentum balance (Eq. 1) were evaluated. As mentioned in the discussion with reference to Orr et al. 2012, the balance is mainly between the Coriolis torque and the wave drag terms. Therefore, including these additional terms would provide interesting information for the S-RIP activity regarding the representation of the residual circulation trends in response to the ozone hole across reanalyses, as well as non-resolved waves and the role of assimilation increments (included in the residual).*

**Reply: The Coriolis torque term has been added to Figure 4, and a comprehensive comparison of the trends in the Coriolis torque and wave-driving (EPFD) terms undertaken, which are shown to be in balance. To describe this in the revised manuscript, the following paragraph has been added to Section 3.2:**

**'*Figure 4 (b,f,j,n) shows analogous results to the EPFD trends but for the Coriolis torque $f\overline{v}^{*}$. The trends in this term are typically similar in magnitude to the trends in the EPFD term but of opposite sign. This is the dominant balance expected under quasi-geostrophic scaling, in part reflecting the fact that both the Coriolis torque on the residual circulation and momentum flux divergence are dominated by the same term: the partial derivative of $F_p^{QG}$ (see Eq. (2)) with respect to pressure, which can be interpreted both as the Coriolis force acting on the net transport of mass by eddies (in the first term on the right hand side of Eq. (1)) and the transport of momentum by eddies, associated with form drag (the second term on the right hand side of Eq. (1); see Vallis (2017), Chapter 10 for further details). Orr et al., (2012) also found that these two terms were of similar magnitude and opposite sign, and that the sum of these two terms agreed well with the zonal wind tendency. Note that differences in the trends in the Coriolis torque were also of similar magnitude but opposite sign to the differences in the EPFD trends.'***

**The role of non-resolved waves and reanalysis increments are discussed in response to comments to Referee #1. Please refer to that response for a detailed reply. However, the main changes to the paper are the following two new additional paragraphs:**

- **In the Data and Methods section, the following new paragraph: '*The key variables examined in Eq. (1) are the trends in the wave forcing (EPFD), and the Coriolis torque $f\overline{v}^{*}$. Our use of the QG TEM approximation for the momentum budget and the lack of complete access to all the relevant data, preclude us from a meaningful analysis of the trends in the residual term in the momentum budget, so therefore this term is not considered. The residual term includes both parameterized gravity wave drag (e.g., Lott and Miller, 1997; Orr et al., 2010) and reanalysis increments in the momentum budget, but also ageostrophic terms and any numerical biases in the model (which therefore cannot be separated as they are included in a single term). The TEM framework is ideal as a diagnostic tool for identifying the dominant balance between the Coriolis torque on the net poleward transport of mass (quantified by the residual circulation) and the transport of momentum by Rossby waves (quantified***

*by the EPFD term), i.e., examining how changes in these two terms relate to changes in the zonal mean wind, which is therefore the focus of this work. On seasonal time-scales, the EPFD and Coriolis torque terms are the leading order balance in the system: momentum transfer in the free atmosphere is controlled dynamically via eddy heat and momentum transfer (Palmer, 1981).'*

- **In the Discussion and Conclusions section, extensive modification to the following paragraph:** *'Consistent with quasi-geostrophic scaling, trends in the Coriolis torque on the residual circulation were nearly in balance with opposite trends in the eddy momentum divergence (EPFD term), as shown in Figure 4. These changes in wave forcing and wave propagation are described by Orr et al. (2012, 2013), as well as other studies such as Hartmann et al. (2000), McLandress et al. (2010, 2011), and Hu et al. (2015). They agree with the temporal evolution of the zonal wind trends, but do not indicate causality. The origin of wind anomalies begins with the slumping of angular momentum surfaces in response to changes in radiative heating by ozone, i.e., the movement of mass to maintain thermal wind balance. The total response depends further on feedback with the resolved eddy forcing, changes in parameterized gravity wave drag, and other ageostrophic terms in the momentum budget. For example, the poleward displacement and intensification of the tropospheric polar front jet in response to the ozone hole is likely to have changed Southern Hemisphere unresolved sources of orographic gravity waves generated by flow impinging on Antarctica (e.g., Hoffmann et al., 2016) and non-orographic gravity waves generated by Southern Ocean storm tracks (e.g., Charron and Manzini, 2002), resulting in changes to the momentum fluxes and drag. However, separating the influence of gravity wave drag, the impact of reanalysis increments, and other residual terms is beyond the scope of the manuscript; as we have used a dataset interpolated to a common grid for the most consistent comparison of the reanalyses, and lack access to all the necessary terms in the residual. This should be the topic of future work. None-the-less, we emphasize the consistency of the dominant balance of the eddy terms with the zonal mean trends, despite the fact that the latter are better constrained by available observations. This internal consistency gives us greater confidence in the overall reanalysis trends.'*

*(2) In the Data Section, it would be helpful to discuss which reanalyses assimilate ozone observations into the model, that is, if the radiative code 'sees' the ozone hole and therefore the described feedbacks are captured in the model, or if in contrast the response is artificially forced by the assimilation increments. This is only briefly mentioned in the discussion (L377-378). Based on your results, is this an important factor?*

**Reply: We have added an additional paragraph (copied below) to the Discussion section, which examines the treatment of ozone in the reanalyses and how it contributes to the radiation calculation in the reanalysis forecast model, and why in the case of the ozone hole it likely has little impact on the final reanalysis temperatures. Sean Davis and Craig Long (NOAA), who are experts on the treatment of ozone in reanalysis (e.g., Davis et al., 2017), advised on the writing of this paragraph. The Discussion section was felt to be a more appropriate location for this information that the Data section, as suggested by the Referee. The new paragraph added to the Discussion section is:**

*'The representation of ozone and its radiative feedback also varies widely between reanalyses and might be an additional factor (Davis et al., 2017). For example, JRA-55, MERRA-2 and CFSR feed the assimilated ozone field to the radiation scheme of the reanalysis forecast model, enabling some ozone-temperature feedback (Davis et al., 2017). However, in ERA5 the ozone field fed to the radiation scheme is based on an ozone climatology, i.e., the impact of ozone depletion associated with the ozone hole on temperature is missing (Hersbach et al., 2020). The primary reason for the assimilation of ozone is that satellite sounder infrared radiances include a contribution from ozone, so knowing the ozone amount helps the radiative transfer code account for that part of the infrared spectrum and thus the thermal contribution (Craig Long, personal communication). However, the assimilated ozone data are generally not available during the long Antarctic polar night, so much of the observed depletion of stratospheric ozone in late winter associated with the ozone hole is not even assimilated (Davis et al., 2017).'*

We have also added an additional sentence to the Data section that refers to the treatment of ozone, which is: '*All the reanalyses assimilate satellite measurements of ozone, although the way that this is treated and the data used varies considerably between reanalysis systems (Davis et al., 2017).*'

*(3) Statistical significance of the trends is not discussed anywhere in the paper. A statistical analysis should be added, not only about the trend significance, but also on the significance of the differences among reanalyses.*

**Reply: Statistical significance testing of the trends has been calculated using the two-sided Student's *t* test, at the 5% significance level. This has been added to Figures 1, 2, 4, 5, 7, 9, and A1. The following sentence has been added to Section 2 explaining this: '*Statistical significance testing of the trends is established using the two-sided Student's t test, with a confidence interval of 5%.*' The trends are generally statistically significant, which is consistent with the results of Section 3.5 and Figure 9. The main text has also been modified to say (when appropriate) that the trends are statistically significant.**

**No significant differences between the reanalyses trends was found, which is expected as the time period is relatively short, noise in the data, and the spread of the trends / differences among the reanalyses are often relatively small (see Figure 10), i.e., the trends are all relatively similar. To reflect this the revised manuscript states that '*Statistical significance testing of the differences in the trends between the reanalyses was also tested, but found to be not significant at the 5% significance level.*' Additionally, the Discussion section includes the new line saying that '*Our results suggest that there is nonetheless a high degree of consistency across the four reanalysis datasets in the representation of the dynamical changes associated with ozone depletion during 1980 to 2001.*'**

*Specific comments:*

*(4) L27: what is the justification for using the quasi-geostrophic version of the EP flux?*

**Reply: The changes in circulation induced by the ozone hole are large-scale, quasi-geostrophic changes. Thus, the quasi-geostrophic formulation of the TEM equations is appropriate. This is discussed in Edmon et al. (1980), which is included as a reference in the manuscript.**

**Edmon, H. J., Hoskins, B. J., and McIntyre, M. E.: Eliassen-Palm Cross Sections for the Troposphere, J. Atmos. Sci., 37, 2600–2616, doi:10.1175/1520-0469(1980)037<2600:EPCSFT>2.0.CO;2, 1980.**

*(5) 133-134: It is fine to show only the fluxes, since they contain the information that you want to discuss. However I find the argument given here not satisfactory, because theta(p) is likely changing substantially due to the lower stratospheric cooling, so it could be important for the divergence (and I assume the full term is used to calculate it).*

**Reply: This point was also picked up by Referee #1. As explained in response to that comment, the sentence arguing why we do not examine the individual EP fluxes has been deleted in the revised manuscript as we agree that it is unsatisfactory, as well as unnecessary.**

*(6) L235: alternating stripes are also seen in the temperature trends (Fig. A1), although at different levels. I wonder if these indicate a more fundamental issue with vertical model levels or assimilation. Are these features described in previous S-RIP papers?*

**Reply: The revised manuscript now mentions that the alternating stripes are also seen in the temperature trends. This is done by adding the following sentence to Section 3.1: '*Although there is also no evidence of vertically alternating differences in the wind trend (Figure 2), alternating negative and positive horizontal stripes are apparent in the temperature trends, albeit it at different levels from the EPFD results (Figure A1).*'**

These features are also evident in the SRIP study by Long et al. (2017), but more properly detailed/discussed in the study by Lu et al. (2014). The Discussion section of the manuscript already includes a paragraph on this aspect (copied below), which explains that they are likely due to either i) issues with temperature increments, or ii) that derivatives are sensitive to interpolation from model levels to standard pressure levels. This paragraph has been modified to include an additional sentence that refers to the horizontal stripes in temperature. The relevant paragraph in the Discussion section is:

*'It is found that, although the circulation trends are generally similar from one reanalysis to the next (with the exception of CFSR), important/large discrepancies in the EPFD trends in the troposphere among the four reanalyses show up as alternating negative and positive horizontal banding (Figure 4), which can be greater than the size of the mean trends across all reanalyses. Lu et al. (2014) suggest that the main contributor for such discrepancies are differences in the vertical derivative of the temperature, which are related to known issues with temperature increments caused by systematic biases in the assimilation of satellite measurements (e.g., Kobayashi et al., 2009). This is consistent with the discrepancies in temperature trends among the four reanalyses, which form vertically alternating negative and positive horizontal bands (Figure A1). An additional factor could also be that derivatives are sensitive to interpolation from model levels to standard pressure levels. However, as there are no vertically alternating differences in the tropospheric wind trend, this suggests that this potential issue is relatively well constrained by analysis increments during data assimilation, while the EPFD is more model dependent. In the lower stratosphere, the trend in EPFD shows little difference among the four reanalyses.'*

*(7) L264-265: "persistent negative values from December to February," except in November at tropopause levels, as you describe briefly in L300-301. Since this is a robust feature, I believe it should be discussed a bit more. Note that there is some spread among reanalyses in the location and strength of these positive momentum flux trends.*

**Reply: This negative trend in eddy momentum flux in the troposphere is one of the key results of the paper. However, I believe that it is already discussed adequately. For example:**

- **This paragraph already contains the additional explanation that '*This occurs at the same time as the poleward displacement of the polar front jet and anomalously strong westerlies in the troposphere (Figures 1 and 2). This negative trend in eddy momentum flux in the troposphere is evident for all four reanalyses products, although JRA-55, MERRA-2, and CFSR have weaker trends than ERA5. Orr et al. (2012) similarly describe strengthened equatorward synoptic-scale wave propagation in the troposphere in response to the ozone hole during the 'decline' and 'decay' stages. They show that this coincides with enhanced baroclinity at the surface (i.e., an increase in upward propagating synoptic-scale waves) at the same latitude as the strengthened polar front jet. This suggests that the circulation trends are the result of the interactions between the zonal-mean flow and the eddies, which maintain anomalies in the polar front jet / tropospheric annular mode. The flux of momentum into the jet (convergence) balances anomalous surface wind stress associated with the shift (see also Hartmann et al., 2000).***

- **The eddy momentum flux from December to February is further examined in Figures 7 and 8.**

- **This result is discussed at length in the fifth paragraph of the Discussion, which focuses on our dynamical understanding of the stratosphere-troposphere system. This paragraph includes the explanation that: '*The strengthening and poleward-displacement of the polar front jet in the troposphere during the 'decline' and 'decay' stages are associated with changes to the synoptic-scale eddy fluxes of momentum and heat that drive the tropospheric annular modes, which is evident by enhanced poleward eddy momentum fluxes into the jet. These changes in wave forcing and wave propagation are described by Orr et al. (2012, 2013), as well as other studies such as Hartmann et al. (2000), McLandress et al. (2010, 2011), and Hu et al. (2015). They agree with the temporal evolution of the zonal wind trends, although this does not necessarily indicate causality.***

*(8) L403-404: but there are stripes in temperature trends (see comment on L235)*

**Reply: This paragraph now includes an additional sentence that refers to the stripes in temperature trends, which is: '*This is consistent with the discrepancies in the temperature trends among the four reanalyses manifesting as vertically alternating negative and positive horizontal stripes*.' See also response to earlier comment.**

Technical corrections:

*(9) CFSR is misspelled (CSFR) in several occasions throughout the paper*

**Reply: This has been corrected throughout the paper. Thanks for spotting this.**

*(10) L164: easterly winds ! easterly wind trends*

**Reply: This change has been made.**

*(11) L242: overbar missing over v'T'*

**Reply: The overbar has been added.**

*(12) L261: add (i.e. poleward momentum transfer) after SH to avoid confusion when*

*comparing to your results.*

**Reply: This change has been made**

*(13) L270: upward ! upward-propagating?*

**Reply: This has been corrected.**

*(14) L272: fluxes ! flux*

**Reply: This has been corrected.**

*(15) L335: exhibit missing an 'i'*

**Reply: This has been corrected.**

*(16) L398: "It is found that," (add comma)*

**Reply: A comma has been added.**

---

## Author Comment (AC3) · 10 Apr 2021

Regarding adding statistical significance to the plots. The reply to reviewers erroneously states that this has been added to Figures 1, 2, 4, 5, 7, 9, and A1. In fact, statistical significance was added to Figures 1-9, A1, and A2.